# Multiplex transcriptional characterizations across diverse bacterial species using cell-free systems

Sung Sun Yim[1,‡] (iD), Nathan I Johns[1,2,†,‡], Jimin Park[1,2], Antonio LC Gomes[3], Ross M McBee[1,4], Miles Richardson[1,2] (iD), Carlotta Ronda[1], Sway P Chen[1,2], David Garenne[5], Vincent Noireaux[5] & Harris H Wang[1,6,*] (iD)

## Abstract

Cell-free expression systems enable rapid prototyping of genetic programs *in vitro*. However, current throughput of cell-free measurements is limited by the use of channel-limited fluorescent readouts. Here, we describe DNA Regulatory element Analysis by cell-Free Transcription and Sequencing (DRAFTS), a rapid and robust *in vitro* approach for multiplexed measurement of transcriptional activities from thousands of regulatory sequences in a single reaction. We employ this method in active cell lysates developed from ten diverse bacterial species. Interspecies analysis of transcriptional profiles from > 1,000 diverse regulatory sequences reveals functional differences in promoter activity that can be quantitatively modeled, providing a rich resource for tuning gene expression in diverse bacterial species. Finally, we examine the transcriptional capacities of dual-species hybrid lysates that can simultaneously harness gene expression properties of multiple organisms. We expect that this cell-free multiplex transcriptional measurement approach will improve genetic part prototyping in new bacterial chassis for synthetic biology.

**Keywords** cell-free expression systems; gene expression; massively parallel reporter assay; synthetic biology; transcription
**Subject Categories** Biotechnology & Synthetic Biology; Microbiology, Virology & Host Pathogen Interaction
**Mol Syst Biol.** (2019) 15: e8875

## Introduction

The cell envelope is a key physical barrier that compartmentalizes cellular functions and insulates biological systems from the external environment. However, this barrier also limits direct access to the genome and other cellular components, complicating genetic engineering efforts and biomolecular characterizations. Cell-free expression systems have long been established as a simplified *in vitro* approach to overcome these challenges by maintaining cellular processes in the absence of an intact cell membrane (Swartz, 2012). Cell-free systems, made up of either individually reconstituted cellular components (Shimizu *et al*, 2001; Wang *et al*, 2012; Villarreal *et al*, 2018) or cell lysates (Jewett *et al*, 2013; Garamella *et al*, 2016), can support a variety of catalytic reactions *in vitro* when supplied with energy sources, cofactors, and ions (Calhoun & Swartz, 2005, 2007; Jewett *et al*, 2008). These cell-free approaches have facilitated the development of therapeutically useful natural products (Dudley *et al*, 2015; Maini *et al*, 2016) and biologics with non-standard amino acids (Martin *et al*, 2018) or chemical moieties otherwise challenging to synthesize (Jaroentomeechai *et al*, 2018). They have also been used to generate viable phage particles directly from purified phage DNA (Shin *et al*, 2012; Garamella *et al*, 2016; Rustad *et al*, 2018) and detect pathogens in emerging diagnostic applications (Pardee *et al*, 2014; Takahashi *et al*, 2018). Recently, cell-free transcription–translation (TXTL; Sun *et al*, 2013; Kwon & Jewett, 2015) has increasingly gained popularity as a prototyping platform for synthetic biology (Hodgman & Jewett, 2012; Takahashi *et al*, 2015). By shortcutting time-consuming cloning and transformation steps, TXTL reactions can directly yield proteins and metabolites straight from DNA that encodes biosynthetic genes, operons, or pathways. TXTL has also been useful for characterizing synthetic gene circuits with multi-layered components and dynamic behaviors (Takahashi *et al*, 2015; Borkowski *et al*, 2018; Marshall *et al*, 2018). However, the correspondence between *in vitro* measurements and actual *in vivo* conditions in live cells remains understudied (Jewett *et al*, 2008; Chappell *et al*, 2013; Siegal-Gaskins *et al*, 2014).

1  Department of Systems Biology, Columbia University, New York, NY, USA
2  Integrated Program in Cellular, Molecular, and Biomedical Studies, Columbia University, New York, NY, USA
3  Department of Immunology, Memorial Sloan Kettering Cancer Center, New York, NY, USA
4  Department of Biological Sciences, Columbia University, New York, NY, USA
5  School of Physics and Astronomy, University of Minnesota, Minneapolis, MN, USA
6  Department of Pathology and Cell Biology, Columbia University, New York, NY, USA
   *Corresponding author. Tel: +1 212 305 1697; E-mail: hw2429@columbia.edu
   ‡These authors contributed equally to this work
   †Present address: Department of Bioengineering, Stanford University, Stanford, CA, USA

Traditional approaches to track RNA or protein levels in TXTL reactions rely on fluorescent reporters that have limited sensitivity and multiplexing capacity. In contrast, deep sequencing offers a vastly improved and less expensive (per construct measured) approach to characterize thousands of genetic designs or variants simultaneously, which can be produced from pooled DNA synthesis or mutagenesis methods. Such massively parallel reporter assays have enabled detailed studies of governing regulatory features of gene expression, such as 5′ untranslated region (UTR) structure and codon usage, directly in cell populations (Kinney *et al*, 2010; Goodman *et al*, 2013; Kosuri *et al*, 2013; Cambray *et al*, 2018). Cell-free studies have also used similar strategies to study transcription from purified RNA polymerases of phage (Patwardhan *et al*, 2009) and *E. coli* (Vvedenskaya *et al*, 2015). Large-scale and rapid analysis of genetic components can thus greatly shorten the design–build–test cycle for synthetic biology.

Recently, a number of TXTL systems have been developed from diverse bacterial species (Kelwick *et al*, 2016; Li *et al*, 2017; Des Soye *et al*, 2018; Moore *et al*, 2018). These TXTL systems allow better design and testing of genetic circuits and metabolic pathways in microbes with unique biochemical capabilities (Moore *et al*, 2017, 2018). Cell-free methods have the potential to more rapidly advance the development of non-model strains that lack characterized genetic parts (i.e., promoters and ribosome-binding sites) into useful chassis for industrial synthetic biology. Additionally, the use of diverse TXTL systems opens the door for systematic assessment of compatibility between circuit components and various chassis. However, the utility of these *in vitro* approaches is dependent on their correspondence to *in vivo* conditions, which are not well-established.

Here, we describe a new deep sequencing-based multiplex strategy to rapidly measure the activities of thousands of regulatory sequences in a single cell lysate reaction, called DNA Regulatory element Analysis by cell-Free Transcription and Sequencing (DRAFTS). We first outline a simple pipeline to rapidly develop and characterize cell-free expression systems in diverse microbes, which we applied to generate active cell lysates from diverse and industrially useful bacteria. Applying DRAFTS to libraries of natural DNA regulatory sequences, we systematically compared transcriptional measurements made *in vitro* in TXTL versus *in vivo* in cell populations. We then analyzed transcriptional patterns across ten diverse bacteria to identify common and diverging transcriptional capacities between different species as well as in "hybrid" cell lysates made from multiple species. This study provides a guiding roadmap to generate diverse cell-free prototyping platforms and use massively parallel reporter assays to expand the regulatory element toolbox for synthetic biology and biomanufacturing applications.

## Results

### Multiplex characterization of 5′ regulatory sequences using DRAFTS

Thanks to recent advances in DNA synthesis, tens to hundreds of thousands of genetic designs can now be cheaply and quickly built in parallel (Kosuri *et al*, 2013; Johns *et al*, 2018). However, these pooled libraries cannot be easily characterized using traditional TXTL reporter assays (e.g., fluorescence), which only support a few channels of simultaneous measurement due to spectral overlap. As a result, individual genetic circuits need to be separately generated and tested, which adds significant hands-on burden and limits throughput, even in 96-well, 384-well, or microfluidic formats (Moore *et al*, 2018; Swank *et al*, 2019). On the other hand, deep sequencing methods are ideally suited for quantitative measurements of many DNA or RNA sequences in pooled reactions. Key considerations when using multiplex measurements are that individual constructs should not cross-interact and must be distinguishable from one another through their sequence or by the use of identifying barcodes. Given these considerations, we designed DRAFTS for quantitative transcriptional analyses of DNA regulatory sequences in pooled TXTL reactions by next-generation sequencing (Fig 1A).

To test whether multiplex transcriptional measurements can be carried out in cell-free expression systems, we first performed pooled *E. coli* TXTL reactions using libraries of uniquely barcoded DNA regulatory parts (ranging from 234 to 29,249 members) derived from our previous study of natural regulatory sequences (Johns *et al*, 2018). These sequences were mined from unidirectional intergenic regions within 184 diverse prokaryotic genomes and consist of the immediate 165 base pairs upstream of target genes, which were synthesized using microarray oligo library synthesis (LeProust *et al*, 2010). Rather than requiring large-scale transformation and propagation of library cell populations, our *in vitro* approach involves incubating single TXTL reactions with library DNA. From each reaction, we performed targeted RNA-seq on reporter mRNAs, which were then enriched by reverse transcription and 3′-end cDNA ligation of sequencing adaptors (Appendix Fig S1). The relative RNA abundance of each construct was normalized to its relative DNA abundance, which was determined by DNA-seq of the input regulatory library, to yield the final quantitative measurement of transcription (see Materials and Methods). Our adaptor ligation-based approach involves amplification of whole 5′ UTRs, enabling identification of transcription start sites (TSSs) by mapping cDNA 3′ ends to reference constructs. Across different regulatory libraries, transcription levels spanned > 5 orders of magnitude and were highly reproducible using the same and separately prepared cell lysates on different days (Fig 1B, Appendix Fig S2A). Similar to previous reports (Johns *et al*, 2018), transcriptional values were consistent across libraries using alternate N-terminal barcodes and downstream genes (sfGFP or mCherry), which supports that gene-specific reverse transcription or barcode identity does not significantly affect our activity measurements (Appendix Fig S2B and C). We observed that relative transcription levels remained constant across different TXTL reaction times (0.5, 1, 2, or 4 h). Lowering input library DNA concentrations from 25 nM to 2.5, 0.25, or 0.025 nM did not introduce significant variability in transcription measurements (Appendix Fig S2D and E). A commercial (myTXTL, Arbor Bioscience) and our laboratory-made *E. coli* lysates also showed highly similar transcriptional profiles, demonstrating reliability of the laboratory-made lysates (Appendix Fig S2F).

TXTL systems are more dilute than the cytoplasm and rates of transcription and translation initiation tend to be lower than what is observed within cells, making it unclear whether *in vitro* measurements will correspond to those made *in vivo* (Jewett & Swartz, 2004). However, one might expect that although absolute gene

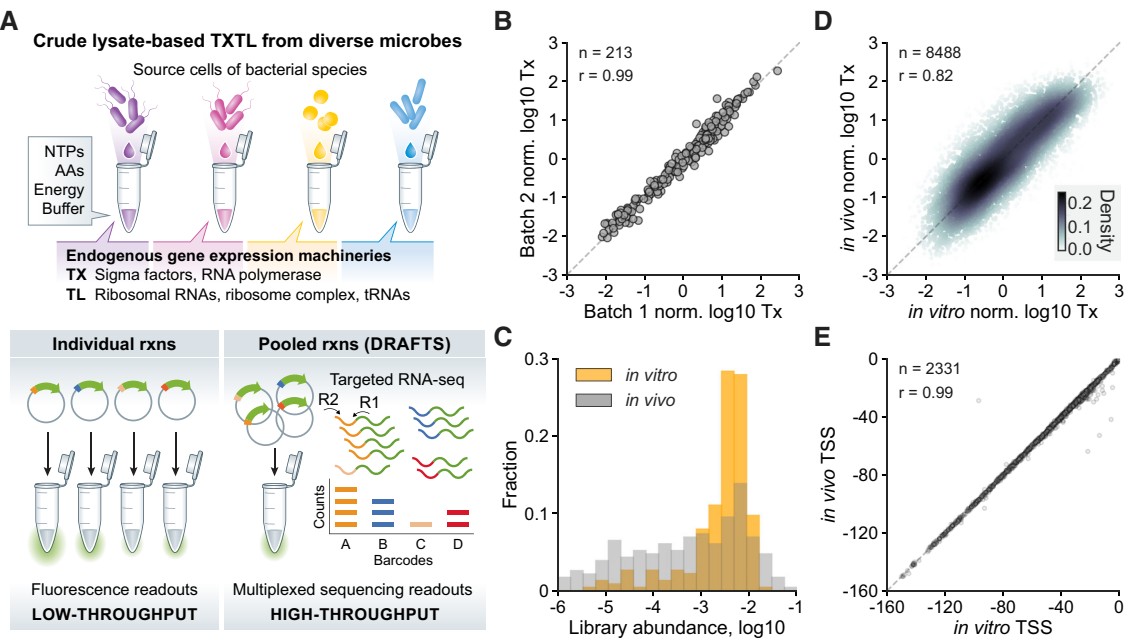

**Figure 1. DNA Regulatory element Analysis by Cell-Free Transcription and Sequencing (DRAFTS).**

A   DRAFTS uses crude cell lysate-based cell-free expression systems to harness source host cell's endogenous gene expression machineries. Compared to conventional single-channel reporters with color or fluorescence readouts, multiplexed sequencing readouts from pooled reactions in DRAFTS scale up the throughput of measurement.

B   Biological replicates of transcriptional profiles (Tx) measured in separately prepared cell-free lysate batches of *E. coli*.

C   Comparison of abundances of each library constructs in *in vivo* and *in vitro* measurements.

D   Correlation between transcriptional profiles (Tx) for regulatory sequence libraries from *in vitro* and *in vivo* measurements in *E. coli*. The color scale indicates the density of data points in a given area of the plot.

E   Correlation between primary TSS calls (in bp from ATG start codon) of regulatory sequences from *in vitro* and *in vivo* measurements in *E. coli*.

Data information: Dashed lines represent $y = x$ in (B), (D), and (E). Sample sizes (*n*) and Pearson correlation coefficients (*r*) are shown in each plot. For normalization, transcription levels in $\log_{10}$ scale were transformed to *Z*-scores. All measurements except (C) are based on two biological replicates.
Source data are available online for this figure.

expression rates are lower, the relative strengths within a promoter library may remain the same. To compare TXTL transcription levels with *in vivo* values, we transformed each library into the same *E. coli* BL21 strain that was used for cell lysate preparation. We observed that the distribution of DNA libraries was much more uniform *in vitro* than *in vivo*, mostly due to the decoupling of gene expression from cell fitness *in vitro*, which improves accuracy by minimizing noisy measurements from low-abundance library members (Fig 1C, Appendix Fig S2G). Importantly, relative transcription levels from *in vitro* and *in vivo* measurements showed a strong correlation, with 94.2% of the regulatory sequence activities within 1-log variation (Fig 1D). We identified transcription start site (TSS) of each regulatory sequence by determining where the start position of RNA-seq reads aligns to reference sequences (Appendix Fig S3A and B, Materials and Methods). TSSs were also highly concordant between *in vitro* and *in vivo* measurements, with 92.6% within 1-bp variation (Fig 1E, Appendix Fig S3C and D). The sequences upstream of TSSs were enriched for sigma70 binding sites, indicating accurate identification of mRNA 5′ positions and enabling compositional analysis of promoter sequences (Appendix Fig S3E and F). Together, these results demonstrate that *in vitro* TXTL reactions provide a reasonable approximation of *in vivo* transcription conditions, enabling high-throughput screening

and quantitative characterization of thousands of regulatory components in a single pooled TXTL reaction.

**A simple pipeline for preparing diverse microbial lysates**

Given the robustness of multiplex transcriptional measurements in *E. coli* TXTL reactions, we sought to generalize this approach by first generating and optimizing cell-free expression systems from other bacterial species. Recent studies have reported the development of cell-free systems from several species other than *E. coli* (Kelwick *et al*, 2016; Li *et al*, 2017; Moore *et al*, 2017, 2018; Wang *et al*, 2018; Wiegand *et al*, 2018). To further expand upon these advances, we established a streamlined experimental pipeline to construct, test, and optimize cell lysates from diverse species using a combination of approaches used by previous studies (Materials and Methods; Sun *et al*, 2013; Kwon & Jewett, 2015). In short, cells are grown in rich media, lysed by sonication, incubated in a run-off reaction, and dialyzed (Fig 2A; Silverman *et al*, 2019). We used the expression of Broccoli, an RNA fluorescence aptamer, to quantify the transcriptional activity of cell lysates (Filonov *et al*, 2014). When bound to the fluorophore DFHBI-1T, Broccoli yields a fluorescence signal that is linearly proportional to its mRNA concentration (Appendix Fig S4C). We systematically optimized reaction buffer

conditions by supplementing with different levels of Mg-glutamate and K-glutamate. The most transcriptionally active buffer conditions were used for all subsequent *in vitro* studies.

In total, we implemented our cell lysate preparation and optimization pipeline on 10 phylogenetically and ecologically diverse

bacterial species including seven Proteobacteria (*E. coli*, *E. fergusonii*, *S. enterica*, *P. agglomerans*, *K. oxytoca*, *P. putida,* and *V. natriegens*), two Firmicutes (*B. subtilis* and *L. lactis*), and one Actinobacteria (*C. glutamicum*; Fig 3A). *E. fergusonii* and *P. agglomerans* strains are novel isolates from the murine gut and

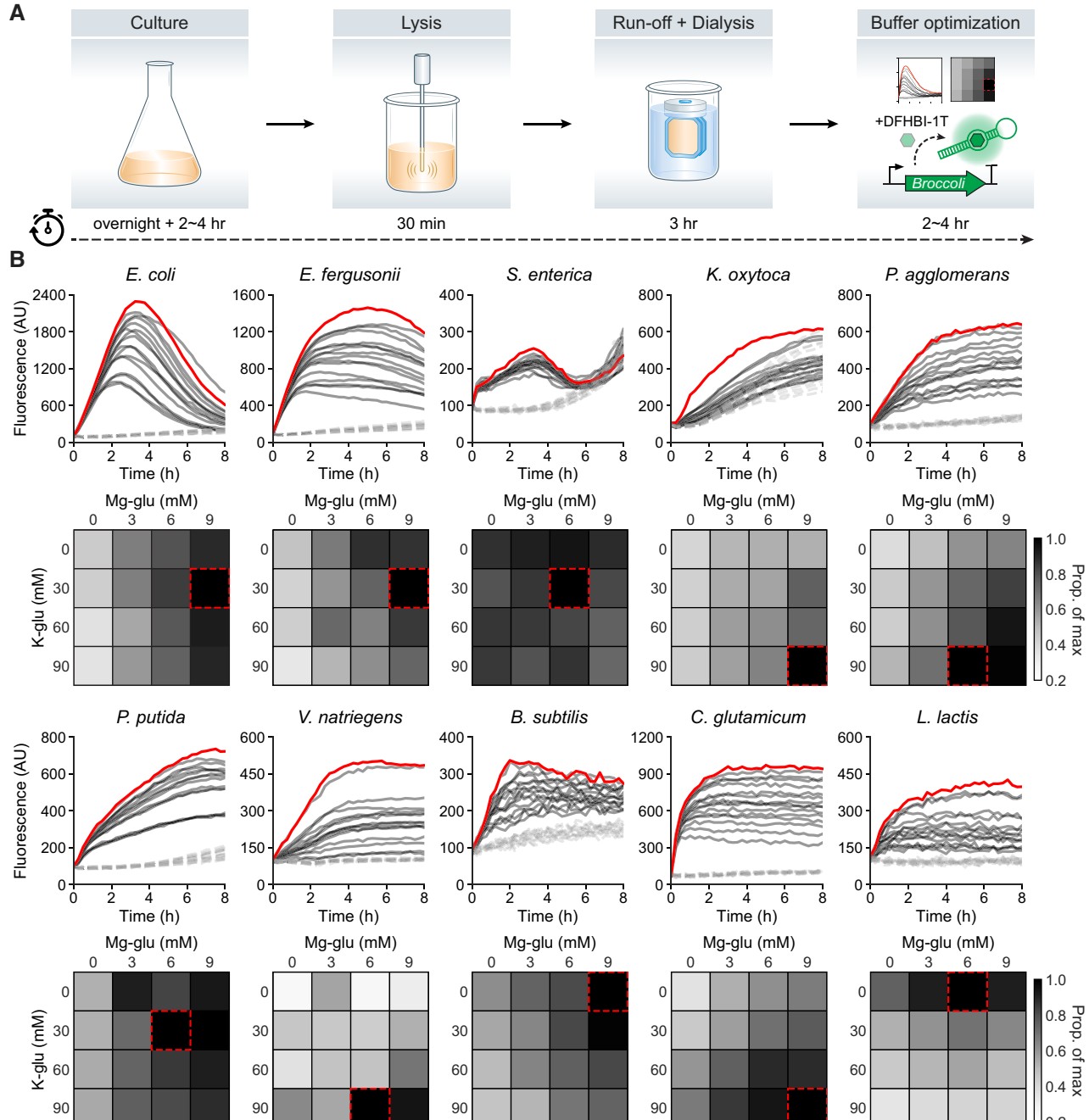

**Figure 2. Development of cell-free expression systems of diverse bacterial species.**

A Schematic diagram of experimental pipeline for preparation and optimization of cell-free expression systems.

B Optimization of transcriptional output using different concentrations of Mg-glutamate and K-glutamate in 10 bacterial cell-free expression systems with Broccoli as a reporter (solid lines, optimal buffer composition shown in red). No DNA template controls are shown as gray dashed lines. 12.5 nM of DNA template was used in all systems, except *L. lactis* (50 nM used).

Source data are available online for this figure.

agricultural waste, respectively, while other strains have been previously described. Each strain was first grown in their optimal growth conditions, and cell lysates were subsequently prepared using the same general protocol (Materials and Methods and Appendix Tables S1–S3). Final cell-free reaction conditions were individually optimized for each species (Fig 2B and Materials and Methods) and quantified to determine transcriptional yields (Appendix Fig S4A). In the *E. coli* lysate, we found that the expression of the Broccoli reporter peaked after ~4 h and then decreased thereafter, consistent with previous reports (Siegal-Gaskins *et al*, 2014). In contrast, *P. agglomerans, V. natriegens*, *C. glutamicum,* and *L. lactis* showed a prolonged Broccoli signal over time with no observed decrease over 8 h. These differences could arise as a result of different stability dynamics of the DNA template or resulting mRNA molecules as well as alternative energy recycling processes for sustaining transcription. While transcription requires a few key proteins and cofactors, translation needs many more components with a far greater degree of coordination between them to function properly. Thus, for cell-free lysate reactions, *in vitro* translation is expected to require significantly more tuning and optimization than *in vitro* transcription. Nonetheless, we quantitatively characterized the translational potential of our 10 cell lysates using an eGFP reporter plasmid and detected measurable levels of protein expression in most lysates except for *L. lactis* (Appendix Fig S4B), although *B. subtilis* and *S. enterica* yielded only small amounts of protein (< 1 ng/μl). These results demonstrate that our cell lysate preparation methods are able to generate active cell-free expression reactions from diverse bacteria with relative ease, provide a foundation for further optimizations of transcription and translation efficiency in cell-free expression systems, and demonstrate their potential for rapid characterization of genetic designs in diverse bacterial species.

**Large-scale transcriptional characterizations in diverse bacteria**

Understanding how microbes utilize regulatory sequences is key for building reliable genetic circuits that experience fewer failures. In previous work, we found that the activity of a regulatory element can vary in different species (Johns *et al*, 2018). A major roadblock in identifying these species-specific regulatory properties is the low efficiency of DNA transformation in many bacteria, which limits multiplex characterization in them. To address this challenge, we first tested a small 234-member regulatory library (RS234; Johns *et al*, 2018) using DRAFTS in seven species with available genetic tools and efficient transformation methods for *in vivo* comparison to obtain multiplex transcriptional data from cell-free systems. These *in vitro* results were compared with *in vivo* transcription levels measured from library-harboring populations of each species harvested at mid-exponential growth phase in rich media (Appendix Tables S1 and S4). Encouragingly, *in vitro* and *in vivo* transcription levels were highly correlated between cell lysates and cell libraries (Pearson's *r* between 0.71 and 0.9, Appendix Fig S5), demonstrating the utility of DRAFTS for accurate multiplex transcriptional measurements in diverse bacteria.

To further scale the throughput of DRAFTS, we mined diverse antibiotic resistance and virulence genes to generate a 1,383-member library of natural regulatory sequences (RS1383; Appendix Fig S1B and Materials and Methods). Since antibiotic resistance and virulence genes are often mobilized between

microbes, we hypothesized that their regulatory regions may exhibit interesting host-range properties. Accordingly, we tested this library in the 10 cell lysates using DRAFTS. By using the same RS1383 library in *in vitro* reactions, we can also minimize any confounding contextual effects of different plasmid backbones, which are otherwise needed for *in vivo* characterizations in different species (Yeung *et al*, 2017). While the RS1383 library was uniformly present following *in vitro* reactions in most lysates, a group of sequences (~13%) was significantly depleted in the *L. lactis* lysate (Appendix Fig S6). Motif analysis revealed a CCNGG motif in these sequences, which corresponded to a known recognition site of a restriction enzyme (ScrFIR) present in the *L. lactis* genome (Twomey *et al*, 1997). We further verified the depletion of these sequences was the result of restriction cleavage in *L. lactis* lysates (Appendix Fig S6E), highlighting the potential interference by bacterial defense systems to confound cell-free studies. We thus removed these sequences from future analyses as they artificially inflated transcription activity calculations.

Many RS1383 regulatory sequences were transcriptionally active in our cell lysates, spanning several orders of magnitude in expression (Appendix Fig S7A). Hierarchical clustering revealed distinct groups of activity levels across phylogenetically diverse bacteria for 421 sequences that were active in all species (Fig 3A and B). For each regulatory element, we performed pairwise comparisons of their transcription levels between species (Fig 3C, Appendix Fig S7B). Pearson correlations of these pairwise comparisons showed varying levels of transcriptional concordance, which when clustered further revealed distinct gram-negative and gram-positive groups (Fig 3C). Principal component analysis of the RS1383 transcription profiles also revealed distinct groups for gram-negative and gram-positive species (Appendix Fig S7C). Pairwise comparisons of 16S rRNA and sigma70 protein sequence similarity showed that more phylogenetically related species tend to share a more similar transcription profile (Fig 3D, Appendix Fig S8). We did not find distinct patterns of expression across different classes of antibiotic resistance genes from which the regulatory sequences were mined (Appendix Fig S9). However, we did observe striking phylogenetic trends, with regulatory sequences derived from the Firmicutes tending to be more highly active overall compared with those from Proteobacteria, which is likely the result of differences in genomic GC contents between these groups (Fig 3E; Johns *et al*, 2018). These findings provide a more informed basis for regulatory element use and design in diverse bacterial species and provide a path for further examination of the sequence features underlying these observed differences.

Accurate models of gene expression have the potential to greatly facilitate the engineering of genetic circuits. While there are accurate tools for predicting translation levels and engineering ribosome-binding sites (Salis *et al*, 2009; Farasat *et al*, 2014), we currently lack similar methods for transcriptional regulatory sequences. To demonstrate the utility of DRAFTS for quantitative modeling of gene expression, we characterized an even larger library of 7,003 regulatory sequences (RS7003; Johns *et al*, 2018) in the 10 cell lysates. We first examined the relationship between 8 regulatory sequence features known to influence promoter activity (i.e., −10/−35 motif strength and interaction (Urtecho *et al*, 2018), spacer sequence, TSS region composition, mRNA 5′-end stability, and GC%; Materials and Methods) and the measured *in vitro*

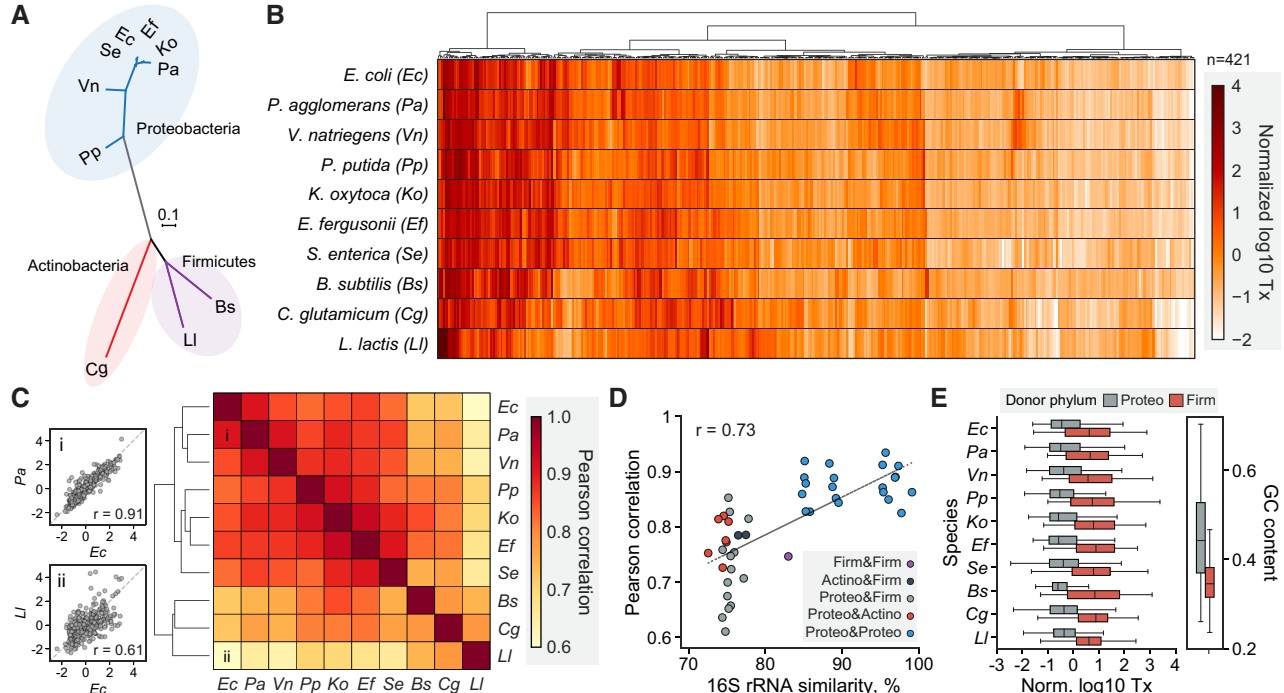

**Figure 3. Comparative functional analysis of regulatory sequences across 10 bacterial species through DRAFTS.**

A Unrooted maximum-likelihood phylogenetic tree of 10 bacterial species used in this study based on multiple sequence alignment of 16S rRNA genes (distance scale of 0.1) using Clustal Omega.

B Transcriptional activities (Tx) of 421 regulatory sequences that were active in all 10 bacterial cell-free expression systems.

C Pairwise comparison of transcriptional profiles of 421 universally active regulatory sequences between bacterial species. Example scatter plots with relatively high and low Pearson correlation in the heat map (marked i and ii) are shown on left.

D Correlation between evolutionary divergences (16S rRNA percent identity) and pairwise Pearson correlation of transcriptional profiles. Dashed line represents linear regression.

E Activity profiles (Tx) of regulatory sequences from donor phyla Proteobacteria and Firmicutes in 10 bacterial species. GC contents of regulatory sequences from the phylogenetic groups are shown on right. Box plots are displaying the interquartile range (IQR) with median values (black line) and whiskers extending to the highest and lowest points within 1.5× the IQR.

Data information: For normalization, transcription levels in $\log_{10}$ scale were transformed to Z-score. All measurements are based on two biological replicates.
Source data are available online for this figure.

expression levels (Fig 4A). In line with previous results (Johns *et al*, 2018), the most predictive features for transcriptional activity across species were the strength of the $-10$ and $-35$ motifs of sigma70 (positively correlated) and the GC content of the regulatory sequence (negatively correlated). We combined all 8 features to generate linear regression models of transcriptional activity for each species and tested the models using repeated random-sampling cross-validation (Fig 4B and Materials and Methods). Transcriptional models based on only 10% of the dataset could explain 27–50% of the variance for these 10 bacterial species (Fig 4C), with 80–98% of predictions falling within 1-log from measured values for transcription activities spanning > 5 orders of magnitude (Appendix Fig S10). In addition, the models were further validated by testing on a separate collection of regulatory sequences (RS1383, Fig 3). The predicted and measured values for the validation were also generally in agreement ranging from 27 to 44% in terms of variance explained (Appendix Fig S11). These results highlight the utility of applying cell-free approaches for deep characterization and modeling of transcriptional processes in diverse bacteria and provide a foundation for studying complex regulatory

mechanisms using *in vitro* sequencing-based approaches (Kinney *et al*, 2010; Rhodius & Mutalik, 2010; Belliveau *et al*, 2018; Cambray *et al*, 2018; Urtecho *et al*, 2018).

## Accessing transcriptional capacities of hybrid lysates through DRAFTS

The diversity of cellular physiologies and metabolisms in the microbial kingdom provides a rich basis to generate cell lysates with unique capacities (Smanski *et al*, 2016). Mixing different cell lysates has been proposed as a way to take advantage of the biochemical features of multiple distinct strains (Fig 5A; Zarate *et al*, 2010; Dudley *et al*, 2015, 2016; Karim & Jewett, 2016). Such an approach might avoid unnecessary cloning and refactoring of biochemical pathways to suit any individual species (Smanski *et al*, 2014). A recent study demonstrated the production of butanediol directly from starch in a hybrid *E. coli*–cyanobacteria (*Synechocystis sp.*), a biotransformation reaction that cannot otherwise be performed in either single-species lysate (Yi *et al*, 2018). Here, the endogenous cyanobacterial starch degrading enzymes increased the amount of

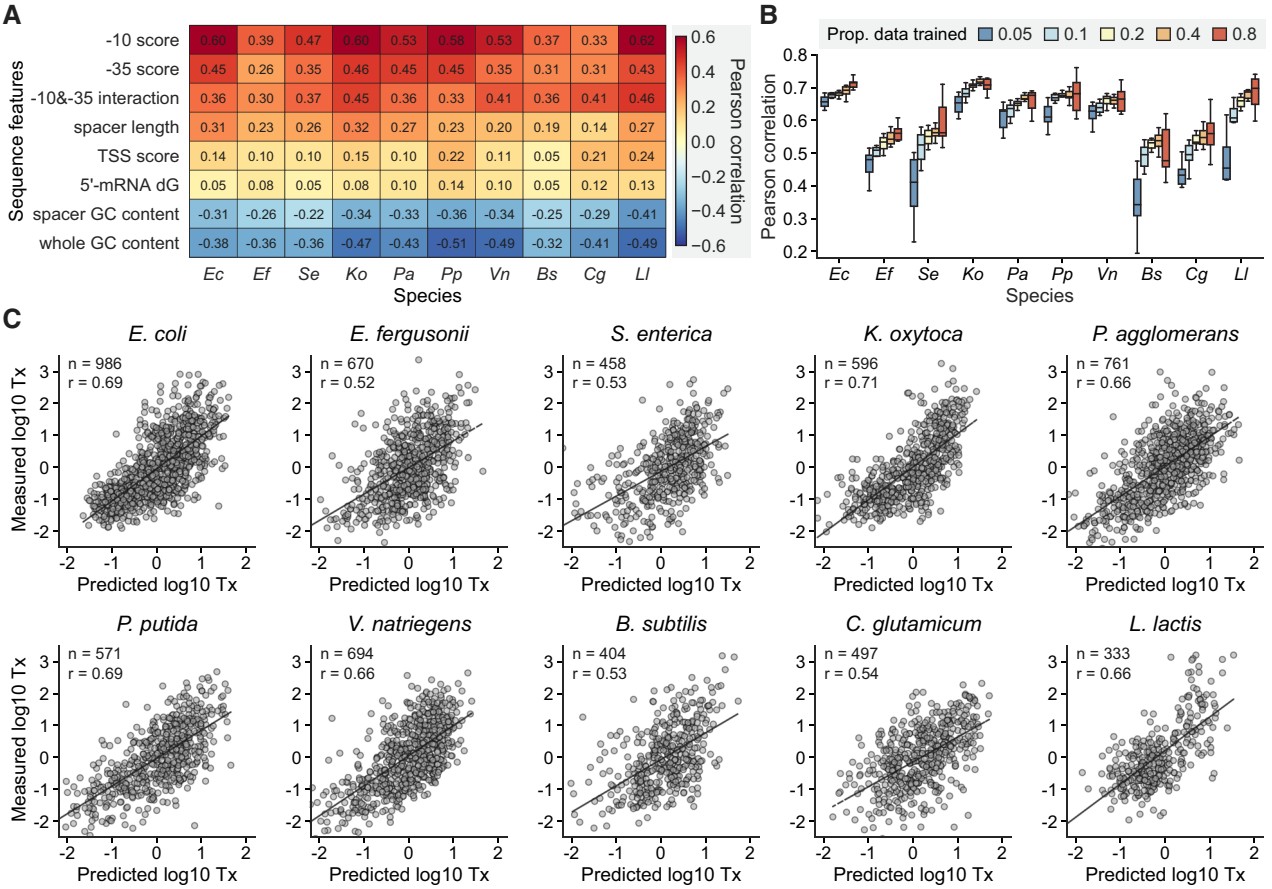

**Figure 4.  Linear regression modeling of transcriptional activation in 10 bacterial species through DRAFTS.**

A   Correlation between sequence features of regulatory sequences and transcriptional activities in each bacterial species.

B   Correlation between predicted and measured transcription levels for each bacterial species with various proportions of data for model training. Data were randomly split for the training and test sets, respectively, and Pearson correlation between predicted and observed transcription levels was computed for 10 times for each proportion. Box plots are displaying the interquartile range (IQR) with median values (black line) and whiskers extending to the highest and lowest points within 1.5× of the IQR.

C   Example linear regression models for transcriptional activation (Tx) in 10 bacterial species using data generated through DRAFTS. Data were randomly split in 10 and 90% for the training and test sets, respectively. Dashed lines represent linear regression. Sample sizes (*n*) and Pearson correlation coefficients (*r*) are shown in each plot.

Data information: For normalization, transcription levels in $\log_{10}$ scale were transformed to *Z*-score. All measurements are based on two biological replicates.

Source data are available online for this figure.

carbon substrate available for biotransformation. Effective harnessing of the biochemical capabilities of hybrid lysates will require knowledge regarding how promoters and translation initiation sequences function in the presence of regulatory machineries from multiple constituent species. To explore the regulatory potential of such systems, we mixed individual cell lysates into 7 dual-species hybrid lysates with equal proportions of each species and then examined their transcriptional and translational activities (Appendix Figs S12 and S13). Interestingly, most hybrid lysates yielded mRNA and protein expression levels that were the average of their single-species counterparts. Importantly, these data suggest that machineries for gene expression from evolutionarily distinct species generally do not negatively interfere with one another in mixed solution, despite the potential for cross-interactions (Sorek *et al*, 2007; Kitahara *et al*, 2012).

To better delineate the transcriptional properties of hybrid lysates, we then applied DRAFTS using the RS7003 library to examine the transcriptional capacity of *E. coli*–*B. subtilis* (*Ec:Bs*) or *E. coli*–*C. glutamicum* (*Ec:Cg*) hybrid lysates generated with different mixing ratios (4:1, 1:1, 1:4). Transcriptional profiles in hybrid lysates were compared to one another as well as non-mixed single species. Principal component analysis revealed transcriptional profiles in individual and hybrid lysates with different mixing ratios that were clearly distinguishable from each other (Fig 5B and C). Interestingly, tuning the ratio of species in the hybrid lysate resulted in transcriptional profiles with more similarity to the majority species (e.g., 1:4 or 4:1) while still retaining some similarity to the minority species (Fig 5D and E). We further performed differential expression analysis on the transcriptional profiles in non-mixed single species to identify promoters with species-selective behavior

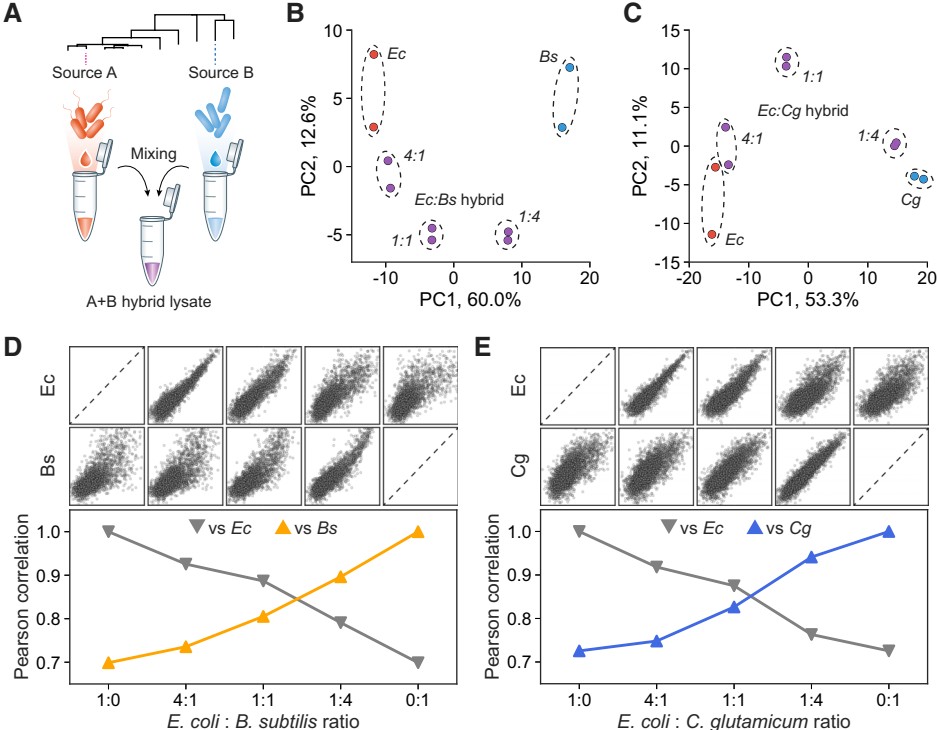

**Figure 5. Transcriptional characterization of dual-species hybrid cell-free systems.**

A       Construction of hybrid systems through cell lysate mixing.

B, C    Principal component analysis of RS7003 transcriptional profile similarity for individual and hybrid lysates with different mixing ratios for (B) *E. coli* + *B. subtilis* and
        (C) *E. coli* + *C. glutamicum* hybrids.

D, E    Pairwise correlation of RS7003 transcriptional profiles between hybrid lysate and single constituent species lysates for (D) *E. coli* + *B. subtilis* and (E)
        *E. coli* + *C. glutamicum* hybrids with different mixing ratios.

Data information: For normalization, transcription levels in $\log_{10}$ scale were transformed to *Z*-score. All measurements are based on two biological replicates.
Source data are available online for this figure.

showing > 10-fold stronger activities in each species (Appendix Fig S14A and D). *B. subtilis* and *C. glutamicum*-selective promoters had lower GC contents in general compared with *E. coli*-selective promoters (Appendix Fig S14B and E), and their species selectivity could be tuned based on the mixing ratios (Appendix Fig S14C and F), a potentially simple strategy for optimizing the synthetic circuits to function within in hybrid lysates. Together, these results demonstrate the construction of functional hybrid lysates from phylogenetically distinct bacteria and also show a high-throughput approach for regulatory element characterization for use in downstream synthetic biology applications using these systems (Zarate *et al*, 2010; Dudley *et al*, 2015; Yi *et al*, 2018).

## Discussion

Cell-free expression systems enable rapid prototyping of natural or synthetic genetic circuits by eliminating time-consuming transformation and cell growth steps. In this study, we developed and characterized cell-free lysates for 10 diverse bacterial species from three phyla (Proteobacteria, Firmicutes, and Actinobacteria), many of which have never been utilized in cell-free expression reactions. Using DRAFTS in our successful lysates, we demonstrated the

accurate multiplexed characterization of thousands of DNA regulatory sequences *in vitro*. This deep sequencing-based approach can quantify transcriptional activities across > 5 orders of magnitude and simultaneously determine transcriptional start sites. *In vitro* transcriptional measurements faithfully recapitulated *in vivo* expression levels, thus avoiding laborious construction of multiple species-specific libraries and transformation into some challenging or recalcitrant species.

We observed increased dissimilarity of transcriptional profiles between phylogenetically more distant species, suggesting a functional evolutionary divergence of transcriptional regulation across the natural microbial biome (Iyer *et al*, 2004). Differences between microbial species could be exploited to build more refined regulatory circuits that function in specific organisms or multiple targeted species (Johns *et al*, 2018). Using these high-throughput datasets, simple linear regression models of transcriptional activation could be trained to predict promoter strengths in a wide array of organisms. However, we expect that more sophisticated computational modeling approaches beyond linear regression used here, such as those employing emerging deep learning frameworks (Urtecho *et al*, 2018), could further improve the generation of accurate, predictable, and useful bacterial transcriptional models and advance the understanding of gene regulation processes in diverse species. Lastly, we

developed multiple hybrid lysates and characterized their gene expression properties, demonstrating tunable species selectivity of regulatory sequences based on mixing ratios. We anticipate that future applications of hybrid lysates that take advantage of the combined metabolic properties of multiple microbes will enable production of useful biomolecules (Yi *et al*, 2018), and may aid in the development of artificial minimal cells with properties of multiple species (Noireaux *et al*, 2011; Johns *et al*, 2016).

While we successfully generated 10 different cell-free lysates that were sufficiently active for deep sequencing measurements of transcription with relative ease, we failed to produce active lysates from other species including *Lactobacillus reuteri*, *L. plantarum*, and *Bacteroides thetaiotaomicron*, which may require further optimizations to address specific properties such as low intracellular pH or strictly anaerobic growth requirements. We expect that strain engineering efforts aimed at deactivating restriction-modification systems or proteases may improve both transcription and translation productivities of cell-free systems made from diverse species of bacteria. Advances in cell lysate preparation, especially for organisms requiring anaerobic or fastidious growth conditions, will further extend these *in vitro* approaches to new areas of bacterial physiology.

Although this study focused on characterizing constitutive promoters, our approach can likely be extended to inducible regulatory systems involving transcription factors or riboswitches (Sharon *et al*, 2012; Rhodius *et al*, 2013; Stanton *et al*, 2014; Chen *et al*, 2018; Forcier *et al*, 2018). In these cases, additional regulatory proteins could be co-expressed with a library of regulatory DNA variants in the same *in vitro* reaction and their ligand-response dynamics could be monitored over time. It might be challenging for cell-free reactions to recapitulate long-term *in vivo* dynamics of genetic circuits relying on turnover of circuit components from degradation or cell division. However, recent advances in cell-free methodologies such as the use of degradation tags or continuous exchange of reaction contents would provide a better mimic of *in vivo* processes while maintaining the simplicity of the *in vitro* format (Niederholtmeyer *et al*, 2015; Garamella *et al*, 2016; Swank *et al*, 2019). Such approaches could also expedite the characterization of promoters controlled by multiple regulators (Belliveau *et al*, 2018) and facilitate their integration into complex regulatory networks (Chen *et al*, 2018; Zong *et al*, 2018). Transcriptional profiles of entire genetic circuits could also be characterized in cell-free reactions using whole-transcriptome library preparation methods, which have been employed *in vivo* to aid debugging of individual components and dynamic circuit behaviors (Gorochowski *et al*, 2017). Sequencing-based quantification methods for translation such as Ribo-seq (Li *et al*, 2014) or polysome profiling (Cambray *et al*, 2018) could enable *in vitro* multiplex characterization of protein synthesis in different bacteria. Finally, we expect multiplexed sequencing and quantification approaches to also improve analysis of fungal (Hodgman & Jewett, 2013), plant (Sawasaki *et al*, 2007), insect (Ezure *et al*, 2006), mammalian (Mikami *et al*, 2006; Martin *et al*, 2017), and other cell-free expression systems derived from eukaryotic organisms to study and engineer increasingly complex layers of gene regulation using synthetic biology.

# Materials and Methods

### Reagents and Tools table

| Reagent/Resource | Reference or source | Identifier or catalog number |
|---|---|---|
| **Experimental models** | | |
| *E. coli* BL21 | New England Biolabs | C2530H |
| *E. fergusonii* | Isolate (murine feces; Ronda *et al*, 2019—PMID: 30643213) | |
| *S. enterica* Serovar Typhi Ty2 | ATCC | 700931 |
| *K. oxytoca* M5A1 | DSM | 7342 |
| *P. agglomerans* | Isolate (agricultural waste) | – |
| *P. putida* KT2440 | ATCC | 47054 |
| *V. natriegens* | ATCC | 14048 |
| *B. subtilis* BD3182 (168 derivative Δ*rok::kan*) | D. Dubnau Lab (Johns *et al*, 2018—PMID: 30052624) | |
| *C. glutamicum* | ATCC | 13032 |
| *L. lactis* | ATCC | 11454 |
| **Recombinant DNA** | | |
| pTOPO-F30-Broccoli | This study | Appendix Table S4 |
| pTXTL-P70a-deGFP | Arbor Bioscience | Cat #502138, Appendix Table S4 |
| RS234 library | This study | Appendix Table S4, Source Data 1 |

**Reagents and Tools table**  (continued)

| Reagent/Resource | Reference or source | Identifier or catalog number |
|---|---|---|
| RS29249 library | Johns *et al*, 2018 —PMID: 30052624 | Appendix Table S4, Source Data 1 |
| RS1383 library | This study | Appendix Table S4, Source Data 3 |
| RS7003 library | This study | Appendix Table S4, Source Data 4 |
| **Oligonucleotides and other sequence-based reagents** | | |
| Primers | This study | Appendix Table S5 |
| **Chemicals, enzymes, and other reagents** | | |
| Media used in this study | | Appendix Table S2 |
| Buffer used in this study | | Appendix Table S2 |
| Cell-free reaction components | | Appendix Table S2 |
| Q5 hot start high-fidelity 2× master mix | New England Biolabs | M0494 |
| SYBR Green I nucleic acid gel stain | Invitrogen | S7563 |
| DNase I (RNase-free) | New England Biolabs | M0303 |
| Maxima H minus reverse transcriptase | Thermo Scientific | EP0753 |
| RiboLock RNase inhibitor | Thermo Scientific | EO0382 |
| RNase H | New England Biolabs | M0297 |
| T4 RNA ligase | New England Biolabs | M0204 |
| DFHBI-1T | Tocris Bioscience | Cat #5610 |
| **Software** | | |
| Python 3.6.0 | https://www.python.org/ (Python software foundation) | |
| BBmerge | https://jgi.doe.gov/data-and-tools/bbtools/bb-tools-user-guide/bbmerge-guide (Bushnell *et al*, 2017 - PMID: 29073143) | |
| NUPACK | http://nupack.org (Zadeh *et al*, 2011 - PMID: 20645303) | |
| MEME | http://meme-suite.org/tools/meme (Bailey, 2002 - PMID: 18792935) | |
| **Other** | | |
| Illumina NextSeq 500/550 | Illumina | |
| NextSeq 500/550 mid output kit v2/v2.5 (150/300-cycles) | Illumina | Cat #20024904/20024905 |
| Illumina MiSeq | Illumina | |
| MiSeq reagent kit v2 (300-cycles) | Illumina | MS-102-2002 |
| CFX96 Touch Real-Time PCR machine | Bio-Rad | |
| Synergy H1 plate reader | BioTek | |
| Q125 sonicator | Qsonica | |
| Slide-A-Lyzer 10k MWCO | Thermo Scientific | Cat #66382 |
| prepGEM bacteria | MicroGEM | PBA0100 |

## Methods and Protocols

### Library construction

Regulatory sequences were mined from databases of antibiotic resistance (Liu & Pop, 2009) and virulence genes (Chen *et al*, 2012) to make the RS1383 library (Appendix Fig S1). Libraries (RS234, RS7003, and RS29249) were derived from our previous study (Johns *et al*, 2018). RS29249 is the full library, RS7003 is a smaller bottlenecked library, and RS234 is a library of isolate Sanger-verified strains that span a wide range of activity levels. Regulatory sequences were mined from unidirectional intergenic regions that were greater than 200 bp in length and consist of the 165 bp preceding target genes. Oligo library design, synthesis, and cloning were performed as previously described (Johns *et al*, 2018). Initial libraries were transformed into *E. coli* MegaX DH10B cells (Invitrogen) and subsequently passaged twice 1:25 in LB + 50 µg/ml carbenicillin. Plasmid DNA was extracted (Zymo Midiprep) and column-purified once more (PureLink, Invitrogen)

to avoid carryover of RNase A into cell-free expression systems. Final concentrations of 10–25 nM were suitable for downstream experiments.

### Lysate preparation

Strain information, media, buffer recipes, and growth conditions used for each species can be found in Appendix. *E. fergusonii* and *P. agglomerans* strains were isolated from mouse feces and hemp feedstock (supplied by Ecovative), respectively. Lysates were generated using similar methods used to produce *E. coli* TXTL systems as follows (Sun *et al*, 2013; Kwon & Jewett, 2015).

- Single colonies of each species were inoculated into 4 ml liquid media and grown 6–8 h.
- 100–500 μl of the culture was transferred to 50 ml of the same liquid media and grown overnight.
- 3.3 ml from the overnight culture was transferred to two flasks containing 330 ml growth medium in 1-l baffled flasks (1:100 dilution) and grown to mid-exponential phase (OD$_{600}$ 1.2–1.6), where the primary sigma factor likely dominates transcriptional control. Host-associated strains were grown at 37°C, while all others were grown at the temperature recommended by the supplier (ATCC, DSMZ) or previous studies (Zhu *et al*, 2008).
- Flasks were rapidly chilled on ice, and cells were centrifuged at 4,300 × *g* for 10 min at 4°C.
- Pellets were washed three times with 50 ml S30A buffer in 50-ml centrifuge tubes.
- The mass of the final pellet was measured, and 0.8 ml (1.2 ml for *L. lactis*) S30A buffer was added per gram of pellet mass and then transferred to Eppendorf tubes in 300 μl aliquots.
- The aliquoted resuspensions were sonicated on ice using a Q125 sonicator (Qsonica) with 3.2-mm probe at 35% amplitude for 4 rounds of 30 s, with 30-s breaks.
- Lysates were clarified by centrifuging at 12,000 × *g* for 10 min at 4°C.
- Supernatant was removed and transferred to 2-ml microtubes with screw cap opened.
- Run-off reactions were performed by incubating clarified lysates at 30°C or 37°C (whichever temperature used for cell growth), 250 rpm for 60–80 min.
- Samples were clarified by centrifuging at 12,000 × *g* for 10 min at 4°C.
- Supernatant was loaded in dialysis cassette (Slide-A-Lyzer 10k MWCO, Thermo Scientific) and dialyzed in S30B buffer for 2–3 h at 4°C.
- Samples were clarified once more by centrifuging at 12,000 × *g* for 10 min at 4°C, then aliquoted, and stored at −80°C.

### Transcriptional optimization

Transcriptional optimization was performed separately for each species. Cell lysates were combined with amino acids, PEG, and energy buffer as previously described (Sun *et al*, 2013), with additional Mg-glutamate and K-glutamate (both from Sigma-Aldrich) at concentrations of 0, 3, 6, or 9 mM and 0, 30, 60, or 90 mM, respectively (total 16 combinations), in a skirted white 96-well PCR plate (Bio-Rad). A plasmid construct (pTOPO-F30-Broccoli) containing strong broad-host-range promoter (Gen_18145) identified by our

previous study (Johns *et al*, 2018), F30-Broccoli, and B0015 terminator was used as a DNA template, and nuclease-free water was used as a negative control. DNA template (2 μl) and 10 mM DFHBI-1T (0.5 μl, Tocris Bioscience) were added to each well immediately before time-course measurements. Fluorescence was tracked for 8 h using a Synergy H1 plate reader (BioTek) at 30°C using excitation and emission wavelengths of 482 and 505 nm, respectively. The concentrations of Mg-glutamate and K-glutamate yielding the highest fluorescence peak were used for future experiments, typically between 1 and 4 h.

### Transcription assay and sequencing library generation

Library measurements were carried out in 10 μl reactions containing 7.5 μl cell-free mixture and 2.5 μl DNA at a final concentration of 10–25 nM. Reactions were incubated at 30°C for 30 min. Each reaction was then split and 2 μl was used for amplification of DNA library and total RNA was extracted from the remaining 5 μl using a Zymo RNA Clean and Concentrator-5 kit, with the remaining volume saved as backups. For *in vivo* measurements, library cultures of all bacterial species were grown in rich media (Appendix Table S1) until mid-exponential phase (OD$_{600}$ ~0.2). RNA was then extracted using RNAsnap (Stead *et al*, 2012), and cells were lysed using prepGEM bacteria kit (MicroGEM) for amplification of input DNA library sequences. Unless otherwise stated, RNA sequencing library was prepared by reverse transcription and common adaptor ligation at 3′-end of the cDNA.

- Gene-specific reverse transcription (Appendix Fig S1A) was carried out for all RNA samples as follows:
  10 μl Total RNA sample (up to 5 μg, DNase I treated and purified)
  1 μl Reverse transcription primer (20 μM)
  1 μl 10 mM dNTPs (Invitrogen)
  2.5 μl Nuclease-free water

- Components were incubated at 65°C for 5 min and chilled on ice for 1 min. The following were then added to the reaction:
  4 μl 5× RT buffer
  0.5 μl RNase inhibitor (RiboLock, Thermo Scientific)
  1 μl Maxima reverse transcriptase (Thermo Scientific)

- The reverse transcription reaction was carried out as follows on a 96-well thermocycler (Bio-Rad):
  42°C for 90 min
  50°C for 2 min
  42°C for 2 min
  Repeat the two steps above for 9 cycles
  85°C for 5 min
  4°C hold

- The completed reaction was incubated with 1 μl RNase H at 37°C for 30 min. Then, the reaction was purified using Zymo DNA Clean and Concentrator-5 kit, and the common adaptor ligation was carried out as follows:
  5 μl cDNA
  2 μl Adaptor (40 μM)

- Components were incubated at 65°C for 5 min and chilled on ice for 1 min. The following were then added to the reaction:
  2 μl T4 RNA ligase buffer
  0.8 μl DMSO
  0.2 μl 100 mM ATP

8.5 µl 50% PEG8000

1.5 µl T4 RNA ligase (New England Biolabs)

- Reactions were incubated overnight at 22°C and purified using Zymo DNA Clean and Concentrator-5 kit.

- Illumina indexes and adaptors were added to both cDNA and input library DNA using a two-step amplification process. All primer sequences are listed in Appendix. Amplifications were performed using the following PCR mixture and cycling:

  10 µl Q5 hot start high-fidelity 2× master mix (New England Biolabs)

  0.2 µl 10× SYBR Green I (Invitrogen)

  1 µl Forward primer (10 µM)

  1 µl Reverse primer (10 µM)

  1 µl 1:10 dilution of library DNA template

  6.8 µl Nuclease-free water (Ambion)

  5 min 98°C Initial denaturation

  10 s 98°C Denaturation

  20 s 62°C Annealing

  30 s 72°C Extension

  Go to step 2 Until end of exponential amplification

  2 min 72°C Final extension

- PCR was performed using a CFX96 Touch Real-Time PCR machine (Bio-Rad). Amplification was stopped as soon as exponential phase ceased, typically around 20 cycles.

- Samples were diluted 1:100 and amplified again using the same protocol using indexing primers for 8–10 cycles.

Samples were then co-purified and examined on a 2% agarose gel to verify correct band sizes of ~200 bp for cDNA and 350 bp for input plasmid DNA libraries.

### Read processing and data analysis

Paired-end reads were merged using BBmerge (Bushnell *et al*, 2017), which simultaneously filters out mismatched reads that can result from poor read quality or multimeric cloning artifacts. Custom code was used to identify the barcode within each read and map the region corresponding to the regulatory sequence to designed library constructs. Transcriptional activities were calculated as done in previous studies (Kosuri *et al*, 2013; Johns *et al*, 2018) by normalizing each construct's individual RNA abundance by its DNA abundance using merged counts from two replicates. The $\log_{10}$-transformed expression levels were converted to *Z*-scores for further comparative analysis across samples. Principal component analysis (PCA) was performed using the PCA function in scikit-learn package by using normalized transcriptional activities of universally active promoters across all samples being compared.

### Determination of transcription start sites and 5′ mRNA stability

To identify the TSSs of regulatory sequences, alignment of the merged RNA reads with the reference sequence was performed after trimming common adaptor sequence with two random bases at the end. Then, we processed TSS calls for each regulatory sequence by an algorithm using kmeans function of scikit-learn package in Python. The algorithm starts with a seed of 16 clusters, and then the number of clusters is reduced if two clusters are found within 10 bp

of each other or if the cluster contains < 1% of all reads. Primary TSS was defined as TSS in which > 70% of TSS calls lie within the cluster with > 200 counts. Other secondary TSSs defined as all the TSSs in which > 10% of TSS calls lie. Free energy of 5′-end mRNA structure was computed using the NUPACK package (http://nupack.org; Zadeh *et al*, 2011). The 5′-end of mRNA was defined by first 50 bp downstream of TSS of each regulatory sequence. Only promoters with primary TSSs were used for this analysis.

### Motif discovery and scoring

The conserved −10 and −35 motifs of regulatory sequences were identified within the 45 bp preceding primary TSSs using MEME software analysis (Bailey, 2002). Spacer sequences between −35 and −10 regions of promoters were extracted during motif scanning and were given penalty scores for suboptimal sizes deviating up to ± 2 bp from the optimal 17 bp. We also examined motifs found within the 8 bp of downstream sequences from primary TSS as this region is important for transcription initiation (Vvedenskaya *et al*, 2015; Yus *et al*, 2017). Unless otherwise stated, top 10% highly active promoters were selected for motif analysis. The motif function in the Biopython package was used to scan the aligned sequences by MEME to obtain motif position weight matrix and match scores for each promoter in the regulatory sequence library.

### Statistical methods

All library measurements were performed in duplicate for both RNA and DNA with replicate values listed in Appendix Table S6. For most analyses, a cutoff of 15 DNA reads was set to minimize noise from low-abundance constructs. All statistics were performed using commonly used Python libraries such as Numpy, Scipy, Pandas, and Seaborn. Linear modeling was performed using the LinearRegression function in scikit-learn package, and cross-validation of the models was performed by randomly splitting the data in 10 and 90% for the training and test sets, respectively, unless otherwise stated.

## Data availability

All data supporting the findings of this study are available within the article and its Appendix and Source Data, or are available from the authors upon request. Sequencing data associated with this study are available at NCBI SRA under PRJNA509603 (https://www.ncbi.nlm.nih.gov/bioproject/PRJNA509603/). Custom code used for data processing is publicly available at the following link: https://github.com/ssyim/DRAFTS.

Expanded View for this article is available online.

### Acknowledgements

We thank members of the Wang Lab for advice and comments on the manuscript. H.H.W. acknowledges funding support from NSF (MCB-1453219), NIH/NIGMS (U01GM110714-01A1), NIH/NIAID (1R01AI132403-01), DARPA (HR0011-17-C-0068), DoD ONR (N00014-15-1-2704), and the Sloan Foundation (FR-2015-65795). V.N. acknowledges funding support from DoD ONR (N00014-13-1-0074) and Human Frontier Science Program (RGP0037/2015). S.S.Y. thanks support from Basic Science Research Program through the National Research Foundation of Korea funded by the Ministry of Education

(NRF-2017R1A6A3A03003401). N.I.J. is supported by an NSF Graduate Research Fellowship (DGE-1644869). We also thank K.J. Jeong (KAIST, Daejeon, Korea) for providing pCES208 plasmid.

## Author contributions

SSY, NIJ, VN, and HHW developed the initial concept. ALCG, CR, SPC, RMM, MR, and DG provided key biological reagents, strains, and code. SSY, NIJ, and JP performed experiments and analyzed the results under the supervision of HHW; NIJ, SSY, and HHW wrote the manuscript with input from all authors.

## Conflict of interest

The authors declare that they have no conflict of interest.

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
