## [Review Process File · Molecular Systems Biology]

Multiplex transcriptional characterizations across diverse bacterial species using cell-free systems

Authors: Sung Sun Yim, Nathan I. Johns, Jimin Park, Antonio L.C. Gomes, Ross M. McBee, Miles Richardson, Carlotta Ronda, Sway P. Chen, David Garenne, Vincent Noireaux and Harris H. Wang.

Review timeline:

Review timeline:	Submission date:	19 th February 2019
	Editorial Decision:	20 th March 2019
	Revision received:	22 nd May 2019
	Editorial Decision:	28 th June 2019
	Revision received:	1 st July 2019
	Accepted:	3 rd July 2019

Editor: Jingyi Hou

Transaction Report:

1st Editorial Decision

20th March 2019

Thank you for submitting your work to Molecular Systems Biology. We have now heard back from the three reviewers who agreed to evaluate your study. As you will see below, the reviewers think that the presented method seems interesting. They raise however a series of concerns, which we would ask you to address in a revision.

I think that the recommendations provided by the reviewers are clear so there is no need to repeat the points listed below. All issues raised by the reviewers need to be convincingly addressed. As you may already know, our editorial policy allows in principle a single round of major revision, so it is essential to provide responses to the reviewers' comments that are as complete as possible. Please feel free to contact me in case you would like to discuss in further detail any of the issues raised by the reviewers.

REFeree REPORTS

Reviewer #1:

In this work the authors present the development of a simple and broadly applicable experimental protocol for cell lysate preparation and demonstrate the use of high-throughput multiplexed sequencing (of both DNA and RNA) to assess the transcriptional activity of large libraries of

promoters. This approach was used to create cell lysates from 10 different bacterial species in which 1000s of regulatory elements mined in a previous study (Johns, N. I. et al., 2018) were tested. The results from this *in vitro* approach were compared to *in vivo* results for a subset of regulatory elements, demonstrating significant correlation (the degree of which varied between species). The experimental results also enabled cross-species and cross-promoter comparisons of transcriptional profiles. This illustrated that species transcriptional profile similarity correlated with their degree of 16S sequence similarity.

Overall the work is novel, interesting and could potentially become a key methodology in synthetic biology. It is clearly shown that it can help accelerate the development of gene regulatory tools in diverse organisms, as well as better understand the design principles for transcriptional regulation. However, I was disappointed that the paper often overstated the scope of its findings, lacked necessary detail given the broad range of experiments and data presented, and failed to properly place itself in the context of the wider work in this area. Furthermore, I wonder if the research might be better framed as a Methods paper for MSB? Most of the biological findings were not particularly insightful (mostly as expected), but the method and its validation are truly interesting and unique and potentially should be expanded on more in the main paper. In light of these points I have the following major comments:

Is any bias introduced by the RT step in the preparation of the sequencing libraries? Some analysis demonstrating this is not the case is necessary and should be included as it could have a major effect on the relative strengths that are measured. Similarly, does the demultiplexing of reads also introduce any bias? Maybe some simulated data could be used to test this second point?

L135-6: It would be useful to include a supplementary plot of *in vivo* gene expression level against *in vivo* library abundance to provide evidence that the broad library abundance distribution *in vivo* is due to gene expression impact upon cell fitness?

L142: It is stated that the TXTL reactions faithfully recapitulate *in vivo* transcription conditions, yet the data clearly does not support this. There is a large variability in measurements >100 fold and there is no way that the *in vitro* reactions display anywhere near the same absolute transcription rates as *in vivo* (purely due to the more highly dilute environment of the gene expression machinery). The authors need to more faithfully present what the data shows (e.g. a good correlation between *in vitro* and *in vivo* transcription, but with a rather large variability of XXX% in many cases). I should stress this doesn't neglect the usefulness of their approach for screening large libraries but will ensure potential users are aware of the actual accuracy they can expect. Related to this, the use of "normalized" Tx in Figure 1d has no real meaning. Units and transformations performed on data should be explicitly stated in the caption. This is seen throughout, and the authors should revise where necessary.

L144: Somewhat related to the previous point, the TXTL reactions will not reproduce the proper coupling between RNAP and ribosomes during protein synthesis due to greatly reduced rates for *in vitro* systems. Will this not affect your measurements? Is it even a big effect? Some details on this are important as a potential limitation of the approach.

L184: What do you mean by the term "flexible"? How is this method any more flexible than an *in vivo* measurement? Please be explicit.

L188: What do the authors mean by the term "regulatory capacity"? I interpret that as the range of ways gene expression can be regulated, but that doesn't fit with what is being described.

L195: More details are required about how transcription levels were measured and the experiments that were performed. This whole section is really important, but details are lacking throughout.

L210: You say that the depletion of specific DNA sequences was "likely" due to restriction cleavage. Why can you not look at the DNA-seq to verify this is the case?

L481: It is stated that correlations between "most" biological replicates was fairly high. How bad were some of the experiments? Details of the libraries should be included in the SI as additional tables/figures.

For the principal component (PC) analyses (Figure 5 & SI Figure 7) it's not clear what data was used to calculate PCs. This should be expanded on and explained in sufficient detail to reproduce the analysis. Please ensure all methods and analysis are clearly explained somewhere in the main text, methods, or links to the SI are given.

Figure 3a: Similar to the previous point, details of sequences used to build the tree should be given. It would be better to compile the tree using maximum-likelihood (e.g. iqtree, <http://www.iqtree.org/>) or bayesian methods. Furthermore, you might consider building a tree to explore dissimilarity between RNA polymerases as the results are specifically related to this molecule & substituting 16S similarity with RNA polymerase similarity in figure 3d. Structural similarity of RNA polymerases of the different species could also be considered.

I also had a few minor comments:

The acronym DRAFTS is memorable, but rather tenuous given that it ignores "cell" in "cell-free". I'd consider either coming up with a better one or merely having a short statement used to describe the method.

You use some very strange words throughout that more international readers may struggle with, e.g. "stymie" (L50). I'd recommend reading through the paper carefully to ensure simple terminology is used where possible to ensure broadest understanding of the work.

The CFE development and characterisation pipeline is omitted from the abstract, it could be included to avoid potentially interested parties overlooking the paper. Furthermore, the paper presents a wealth of data on function of promoter sequences in different species, by making this clear in the abstract the sequence information may be usable in future by parties working with the species characterised

L83: Typo, should read "chassis"

L89, 96, 184: Throughout the document "new" should be clarified or replaced with something along the lines of "organisms from which a cell-free lysate has not previously been prepared" to refer to microbes which haven't had a cell-free lysate prepared before; the microbes are not new.

L105: "support single-channel of measurement" -> "support single-channel measurements"

L109: You talk about assumptions in multiplexing but neglect the fact that every sequence must be distinguishable, e.g. via a barcode. This should be added.

L110: Some readers may misunderstand species as in different species of organism, perhaps clarify that nucleic acid species is being referred to.

L132: Include reference to Supplementary Figure 2 which shows the similarity.

L136: You should be explicit about why obtaining uniform numbers of DNA reads across the library helps improve multiplexed measurements. This sort of omission occurs throughout. The authors should carefully check that they backup their statements with proper explanations so that it is not assumed the reader understands their rationale.

L138: You don't explain how the TSS was found. A short description would help greatly in understanding the process, with the main details in the methods.

L150: "several non-E. coli species" -> "several species other than E. coli".

L156: Refer the reader to SI figure 4c for the broccoli standard curve.

L159-164: Indicate Figure 3a which shows unrooted phylogenetic tree of species.

L230, 247: References supporting these claims need to be added.

L241: I was shocked that the biggest study to date in using sequencing to assess mechanistic aspects of gene regulation (translation initiation and elongation) was not cited, i.e. Cambray et al. Nat Biotech, 2018. Is there a reason for this?

L291: "increased regulatory dissimilarity"? Consider rewording.

L324: If the authors would like to introduce potential applications with genetic circuits it is essential that they outline the major issues/challenges using these in bulk reactions. Many circuits with intricate dynamics rely on turn-over of mRNAs and proteins that may not happen in vitro. Such issues and potential solutions should be mentioned.

L337: Rudimentary details of regulatory element mining process would be useful for the reader.

L350: There are no details of how the temperature for each strain was selected for growth. This should be explained in more detail.

L357: "4,300xg" -> "4,300 x g"

L451: log10 should use subscript.

L452: It would be useful to show the distributions of expression levels and assess that they are normally distributed (a necessary condition for using Z-scores).

L486: Was any cross-validation done when carrying out the LinearRegression function. If not, why?

Figure 1: $x = y$ lines could be added to panels b and d, plus the use of a density plot would help better visualize how dense the central regions are.

SI Figure 5: 95% confidence interval lines along line of best fit would be beneficial.

I very much enjoyed reading the paper, believe it is an important methodology, and hope these comments are useful to the authors in improving their manuscript.

Reviewer #2:

The authors present a multiplexed, cell-free approach to characterizing DNA regulatory elements using Illumina sequencing. Specifically, their approach uses cell-free transcription and translation with pooled DNA containing the elements to be tested. Then, they reverse transcribe RNA produced in this reaction to obtain cDNA which is then sequenced to obtain abundance of transcripts from each DNA element. This approach is termed DRAFTS. They validate this approach by comparing their results from a DNA library of regulatory elements with a cellular dataset from a previous study. These data align well. The authors then make lysates out of 10 strains of bacteria from evolutionary diverse backgrounds, many of which no previous report of lysate preparation has been published. They perform a simple magnesium and potassium optimization of each lysate using transcription as an output. They take these 10 lysates and test a library of 421 regulatory sequences through DRAFTS and compare their transcriptional outputs in relation to their evolutionary distance from one another. They also find GC-content of these regulatory sequence to be important. They then use this data to generate a somewhat predictive model to describe transcription in each lysate. They use that model to predict transcription activation for 7,003 sequences and use DRAFTS to test these predictions. Their models are roughly predictive ($r=0.52-0.71$). Then, they mix lysates from divergent species together and show that by varying the ratio of each species' lysate they can modulate transcription to be more or less like each individual strain suggesting that the impact of each species might be additive. This work represents an important step forward for cell-free systems in the context of prototyping in that it brings together next generation sequencing and multiplexed reactions. This is the biggest innovation and sets an important precedent. There are clear applications toward prototyping DNA elements for in vivo production of transcripts and proteins in a diverse array of bacteria though this work did not do a sufficient job in comparing/showing this.

This work will be of interest to the synthetic biology community, is timely, and well written. However, some elements lessened my enthusiasm and those significant concerns are described below.

Major Points:

- While the work is impressive, the work lacks sufficient comparison to cellular data. The authors state that one of the main problems they are trying to address is the lack of connection and understanding between in vitro measurements and in vivo conditions. I am not sure if this paper changes our understanding since the focus is on *E. coli*. While the cell-free experiments in *E. coli* lysates align nicely with *E. coli* cells, which is expected based on previously published work, there is no in vivo data shown for any of the other nine species. This needs to be done to make their claims and to enhance the impact of the paper.
- The idea that the authors have a unique or novel approach to lysate prep is not supported. Their development of cell-free expression systems of diverse bacterial species appears to follow within slight variations of previously published methods of optimization, including works from the authors (Noireaux) and others such as ref 18, ref 36, ref 39. The authors should put their work in better and more accurate context since their pipeline is just a variation of what has been reported before.
- Can the authors comment more on why the *L. Lactis*, *B. subtilis*, and *S. enterica* extracts didn't work well? Or that *Lactobacillus reuteri*, *L. plantarum*, and *Bacteroides thetaiotaomicron* didn't? This would be very informative for the field since many cell-free systems have been recently reported from bacterial species using slight variations of the lysate preparation methods used in this paper and those previously published. Also, did they try and not report broader salt conditions to try to get it to work?
- The comparative characterization of transcription in diverse bacterial lysates is observational and it is unclear what new knowledge is gained by this. For publication in a journal like MSB, rather than a specialty journal, it seems that there should be new insights gained.
- It is unclear how their predictive models differ from previously published regulatory sequence calculators, and whether their prediction are better than the state-of-the-art (<http://salislab.net>). The authors should quantitatively demonstrate that their method is on par or better than the Salis method and comment on this
- In addition, the hybrid cell-free systems is a nice proof-of-concept but the importance or even understand gained from this is unclear. A more thorough analysis of systems biology would be helpful.
- Also with respect to the hybrid systems, I am still trying to understand the justification of "Effectively harnessing the biochemical capabilities of hybrid lysates will require knowledge regarding how promoters and translation initiation sequences function in the presence of regulatory machineries from constituent species." It seems this wouldn't be required if you just used enzymes from the cells produced before making the lysates. Can the authors better explain the need for the non-expert?

Minor Points of interest:

- Line 48, isn't cell membrane a better term than cell envelope? I wasn't sure technically speaking what an envelope is.
- In supplemental figure 5, only 7 lysates are tested with no comment or justification. Why didn't the authors test the 234 member library in all 10 bacterial strains? This needs to be better described.
- Line 210, the hypothesis here is that a specific RE, ScrFIR, is present in *L. lactis*. This is a testable hypothesis. It would strengthen the manuscript to test this hypothesis experimentally, validate the claim, and close the loop.
- In supplemental figure 10 what are the units on the y-axis? The authors should covert to microM or nanoM.
- Lines 146-186, the authors should tone down the language of this being a new pipeline of making lysates. The pipeline presented in this manuscript matches previous reports of lysate preparation and optimization and is not new.
- Line 180, is your detection sensitive enough to say that your *B. subtilis* and *S. enterica* have any protein produced?

Reviewer #3:

The authors develop and optimize TX-TL assays using the cell lysates from a diverse collection of 10 bacterial strains that support suitably high levels of transcription. They combine oligopool-based library cloning and next-generation sequencing (DNA-Seq, RNA-Seq) to characterize the transcription rates from hundreds to thousands of natural promoters in TX-TL across the 10 species' lysates. From this data, they identify several promoter sequence features that correlate well with higher transcription rates, including -10 & -35 hexamer sequences, GC content, and spacer length. They also re-confirm that AT-rich promoters are generally transcribed better across many organisms, supporting a finding of the authors' previous work. Overall, the key novelty of the work is the development of TX-TL assays from the cell lysates of diverse organisms, and their utilization to measure promoter transcription rates. However, the data analysis overall was fairly underwhelming; we do not learn a great more from the authors' analysis, compared to what is already known in the literature about promoter specificity.

Specific Comments:

1. The authors use the phrase "predictively modeled" or "predictive modeling" in several places in the abstract and manuscript. However, I see only statistical linear regression models that attempt to find correlative relationships between selected features and outcomes, using standard cross-validation techniques that split one data-set into training and test sets. These are not considered predictions in normal usage of the word (ie, forward-engineering) on unseen data. Moreover, the authors do not test the error of what they call a "prediction", for example, by directly comparing a predicted to measured transcription rate. All metrics are global and statistical (e.g. Pearson's R), and it is well-known that Pearson's R metric has several known defects, which make it unsuitable for testing the "predictive" accuracy of any model by itself. Judging from their analysis, I don't think the authors have any intention that their model would be used to predict a promoter's transcription rate across organisms, and yet they use the word "predictive modeling" several times. The authors either need to remove any mention of "predictive modeling" from the manuscript or substantially support their claims by using their model to predict the transcription rates of a diverse collection of new (perhaps synthetic) promoters, followed by construction and TX-TL characterization across their selected organisms. It's important here to see the full error distribution of any model that attempts to make predictions.
2. Given the different organisms' differences in TX-TL transcriptional output and distinctly different mRNA dynamics across the organisms, it's not clear how the authors are deriving their "Observed Log₁₀ Tx" values. The authors need to very clearly describe (in the main text) how they extract a single number from their characterized mRNA level dynamics across the different organisms.
3. In Figure 1C, the y-axis is labeled 'Density' (ie, probability density). However, it's clear that the probability distributions shown are not correctly normalized from their raw counts. If a correct probability distribution is integrated across its interval, it must sum to one, which is not true here. To be more clear and correct, the authors should label the y-axis as 'Probability' and divide all densities such that their sum across the interval is equal to one. This will alter the heights of the probability distributions, but not their shapes, which means the authors' conclusions remain supported.
4. Related to Figure 2b, the authors mention several possibilities for why the dynamics of mRNA levels vary across TX-TL reactions made using different organism cell lysates. However, the authors do not mention if the organisms could have been grown in conditions where their sigma factor levels are distinctly different. For example, how do we know if the sigma⁷⁰ vs. sigma^S (homolog) levels in *S. enterica* cell lysate are the same as in the *E. coli* lysate? These differences would have a large effect on promoter sequence specificity in TX-TL reactions. What steps did the authors take to "regularize" the growth conditions across all organisms so that their sigma factor levels are the same, keeping in mind that exponential growth conditions in one organism are distinct from another?

Point-by-Point Response to Reviewer Comments

We thank the reviewers for their thoughtful comments and suggestions to improve the manuscript. Accordingly, the specific reviewer comments (R#.#) are in black and authors' responses (A#.#) are in blue.

Reviewer #1:

In this work the authors present the development of a simple and broadly applicable experimental protocol for cell lysate preparation and demonstrate the use of high-throughput multiplexed sequencing (of both DNA and RNA) to assess the transcriptional activity of large libraries of promoters. This approach was used to create cell lysates from 10 different bacterial species in which 1000s of regulatory elements mined in a previous study (Johns, N. I. et al., 2018) were tested. The results from this in vitro approach were compared to in vivo results for a subset of regulatory elements, demonstrating significant correlation (the degree of which varied between species). The experimental results also enabled cross-species and cross-promoter comparisons of transcriptional profiles. This illustrated that species transcriptional profile similarity correlated with their degree of 16S sequence similarity.

Overall the work is novel, interesting and could potentially become a key methodology in synthetic biology. It is clearly shown that it can help accelerate the development of gene regulatory tools in diverse organisms, as well as better understand the design principles for transcriptional regulation. However, I was disappointed that the paper often overstated the scope of its findings, lacked necessary detail given the broad range of experiments and data presented, and failed to properly place itself in the context of the wider work in this area. Furthermore, I wonder if the research might be better framed as a Methods paper for MSB? Most of the biological findings were not particularly insightful (mostly as expected), but the method and its validation are truly interesting and unique and potentially should be expanded on more in the main paper. In light of these points I have the following major comments:

Major comments

R1.1: Is any bias introduced by the RT step in the preparation of the sequencing libraries? Some analysis demonstrating this is not the case is necessary and should be included as it could have a major effect on the relative strengths that are measured. Similarly, does the demultiplexing of reads also introduce any bias? Maybe some simulated data could be used to test this second point?

A1.1: Our reverse transcription uses a gene-specific primer that is complementary to the GFP or mCherry reporter gene sequence and the following PCR reaction amplifies whole 5' UTRs which include our barcode sequence (**Appendix Fig S1, Appendix Fig S2c**). A random priming approach could introduce biases as the reviewer suggests. On the other hand, the use of a primer within the constant reporter construct avoids these issues and is the gold-standard for multiplex reporter assay and we have cited several studies using similar designs (Kosuri et al., 2013; Johns et al., 2018; Urtecho et al., 2019). We also use a highly processive reverse transcriptase (Maxima H minus reverse transcriptase, Thermo Scientific) that is capable of producing long cDNAs, even in the presence of secondary structure. A very similar targeted RNA-seq methodology was validated using qPCR previously (Bagnoli et al., 2018; Daniel et al., 2019). We do not expect there to be any issues with demultiplexing as both sample indexes and construct barcodes are sufficiently distinct from one another (minimum Hamming distance of 3) to avoid mapping errors. In addition, alternative barcodes or reporter genes for the same library constructs yielded consistent transcription levels as shown in **Appendix Fig S2b,c**, which also supports that demultiplexing of reads or RT step does not introduce bias. Nonetheless, we thank the reviewer for pointing out these points and to clarify them to the reader, we have revised the text with more details (line 109-111, 132-133, 443).

References

- Kosuri et al., Composability of regulatory sequences controlling transcription and translation in *Escherichia coli*. *Proc Natl Acad Sci USA* 110, 14024-14029, (2013).
- Johns et al., Metagenomic mining of regulatory elements enables programmable species-selective gene expression. *Nat Methods* 15, 323-329, (2018).
- Urtecho et al., Systematic dissection of sequence elements controlling sigma70 promoters using a genomically encoded multiplexed reporter assay in *Escherichia coli*. *Biochemistry* 58, 1539-1551, (2019).
- Bagnoli et al., Sensitive and powerful single-cell RNA sequencing using mcSCR-seq. *Nat Commun* 9, 2937, (2018).
- Daniel et al., Performance comparison of reverse transcriptases for single-cell studies. *bioRxiv* <https://doi.org/10.1101/629097>, (2019)

R1.2: L135-6: It would be useful to include a supplementary plot of *in vivo* gene expression level against *in vivo* library abundance to provide evidence that the broad library abundance distribution *in vivo* is due to gene expression impact upon cell fitness?

A1.2: We thank the reviewer for the useful comment. We have now included this plot (**Appendix Fig S2g**), which confirms the negative relationship between expression level and *in vivo* library abundance.

R1.3: L142: It is stated that the TXTL reactions faithfully recapitulate *in vivo* transcription conditions, yet the data clearly does not support this. There is a large variability in measurements >100 fold and there is no way that the *in vitro* reactions display anywhere near the same absolute transcription rates as *in vivo* (purely due to the more highly dilute environment of the gene expression machinery). The authors need to more faithfully present what the data shows (e.g. a good correlation between *in vitro* and *in vivo* transcription, but with a rather large variability of XXX% in many cases). I should stress this doesn't neglect the usefulness of their approach for screening large libraries but will ensure potential users are aware of the actual accuracy they can expect. Related to this, the use of "normalized" Tx in Figure 1d has no real meaning. Units and transformations performed on data should be explicitly stated in the caption. This is seen throughout, and the authors should revise where necessary.

A1.3: We have clarified in the text (line 149) that the comparisons between *in vivo* and *in vitro* transcriptional measurements are relative values as we do not directly measure absolute rates of transcription. We also revised the text to include that that 94.2% of the regulatory sequence activities measured in cell-free reactions fell within 1-log of *in vivo* values and transcription start sites (TSSs) identified in cell-free reactions were also highly concordant with TSSs identified *in vivo* with 92.6% within 1-bp variation (line 149-154). We also added density information to the scatter plot in **Fig 1d** to better display the variability in the comparison being made. We have also made clarifications to our figure axis labels and captions to explicitly state that we used the Z-score conversions of log₁₀ transformed data to facilitate comparisons between samples, as is common in many transcriptomic studies.

R1.4: L144: Somewhat related to the previous point, the TXTL reactions will not reproduce the proper coupling between RNAP and ribosomes during protein synthesis due to greatly reduced rates for *in vitro* systems. Will this not affect your measurements? Is it even a big effect? Some details on this are important as a potential limitation of the approach.

A1.4: The reviewer is correct that the cell-free systems are in lower concentration and have reduced rates of gene expression compared to cellular environment *in vivo* and therefore the synchronization of transcription and translation may also be altered in cell-free systems

(Jewett and Swartz, 2004). However, previous studies using libraries of promoters and ribosome binding sites have suggested that transcription and translation can be tuned mostly independent of one another (Kosuri et al., 2013), and more importantly, the relative transcription levels among the library pool were found to be highly correlated between *in vivo* and *in vitro* measurements regardless of absolute transcriptional and translational yields of cell-free systems in our data (**Fig 1d, Appendix Fig S4,S5**). Taking the reviewer's comment into account, we have revised the text (line 140-144) to include the considerations regarding intrinsic differences between *in vitro* and *in vivo* environments.

References

- Jewett and Swartz, Mimicking the *Escherichia coli* cytoplasmic environment activates long-lived and efficient cell-free protein synthesis. *Biotechnol Bioeng* 86, 19-26, (2004)
- Kosuri et al., Composability of regulatory sequences controlling transcription and translation in *Escherichia coli*. *Proc Natl Acad Sci USA* 110, 14024-14029, (2013).

R1.5: L184: What do you mean by the term "flexible"? How is this method any more flexible than an *in vivo* measurement? Please be explicit.

A1.5: By flexible we meant that the protocol does not require species-specific vectors and thus avoids laborious cloning and library transformation steps, making it simpler than conventional *in vivo* methods. These benefits make our approach an attractive option for studying systems of higher complexity. However, we take the reviewer's point and have decided to simplify the sentence for clarity as the benefits of our methodology are adequately explained elsewhere in the manuscript.

R1.6: L188: What do the authors mean by the term "regulatory capacity"? I interpret that as the range of ways gene expression can be regulated, but that doesn't fit with what is being described.

A1.6: We have made clarifications to this sentence to improve clarity.

R1.7: L195: More details are required about how transcription levels were measured and the experiments that were performed. This whole section is really important, but details are lacking throughout.

A1.7: We thank the reviewer for pointing this out. To address this, we have added more experimental details in the body of the text that we measured the *in vivo* transcriptional levels from library-harboring populations of each species harvested at mid-exponential growth phase in rich media (**Appendix Table S1,S4**), and also in the Method section (line 210-212, 437-440).

R1.8: L210: You say that the depletion of specific DNA sequences was "likely" due to restriction cleavage. Why can you not look at the DNA-seq to verify this is the case?

A1.8: In **Appendix Fig S6**, we are deriving the depletion of the regulatory sequences with CCNGG recognition site of the restriction enzyme (ScrFIR) in *L. lactis* using DNA-seq data. To further demonstrate the restriction enzyme activity in this revision, we additionally examined the stability of a DNA fragment containing the putative recognition site (CCTGG) and quantified its degradation compared to a control DNA fragment (the same sequence but without the motif) in *L. lactis* lysate using qPCR and have added this new result as **Appendix Fig S6e**.

R1.9: L481: It is stated that correlations between "most" biological replicates was fairly high. How bad were some of the experiments? Details of the libraries should be included in the SI as additional tables/figures.

A1.9: As suggested by the reviewer, we now have included the replicate correlation values with our raw data in **Appendix Table S6** and revised the text (line 533-534) accordingly. Here we summarized the table as a histogram on the right. In addition to *L. lactis* lysate with the strong restriction enzyme activity (**Appendix Fig S6e**), *S. enterica* lysate also showed relatively lower reproducibility which is likely due to suboptimal gene expression efficiency (**Appendix Fig S4a,b**). But in general, ~97% of the experiments were with robust reproducibility (Pearson $r > 0.8$) and ~72% were with Pearson $r > 0.9$ between biological replicates. It should also be noted that for all species, most of the variability for biological replicates occurs in low expression promoters.

R1.10: For the principal component (PC) analyses (Figure 5 & SI Figure 7) it's not clear what data was used to calculate PCs. This should be expanded on and explained in sufficient detail to reproduce the analysis. Please ensure all methods and analysis are clearly explained somewhere in the main text, methods, or links to the SI are given.

A1.10: The principal components are computed using all transcriptional activities from universally-active promoters in cell-free systems of each species or hybrids being compared by using PCA function in scikit-learn package. To clarify this point, we have added these details about principal component analyses in the Method section (line 504-506).

R1.11: Figure 3a: Similar to the previous point, details of sequences used to build the tree should be given. It would be better to compile the tree using maximum-likelihood (e.g. iqtree, <http://www.iqtree.org/>) or bayesian methods. Furthermore, you might consider building a tree to explore dissimilarity between RNA polymerases as the results are specifically related to this molecule & substituting 16S similarity with RNA polymerase similarity in figure 3d. Structural similarity of RNA polymerases of the different species could also be considered.

A1.11: We thank the reviewer for this suggestion. We have now provided the 16S rRNA sequences in **Source Data 3** and have replotted **Fig 3a** using a maximum-likelihood approach, which yielded the same relationships as before. More importantly, we also examined the relationship between sigma70 protein divergence and transcriptional profile similarity as suggested by the reviewer. Indeed, we observed a similarly strong evolutionary trend as we see in **Fig 3d** with 16S rRNA similarity and have included this as a new **Appendix Fig S8**. Details can be found in the figure and **Source Data 3**. We agree to the reviewer that it would also be interesting to explore how structural evolution of RNA polymerases could impact activation of natural and synthetic promoters. Unfortunately, there is little publicly available structural data beyond what is available for *E. coli* and this would be beyond the scope of this paper.

Minor comments

R1.12: The acronym DRAFTS is memorable, but rather tenuous given that it ignores "cell" in "cell-free". I'd consider either coming up with a better one or merely having a short statement used to describe the method.

A1.12: We would like to keep the acronym to maintain text flow in the manuscript and, as the reviewer notes increased memorability of the approach.

R1.13: You use some very strange words throughout that more international readers may struggle with, e.g. "stymie" (L50). I'd recommend reading through the paper carefully to ensure simple terminology is used where possible to ensure broadest understanding of the work.

A1.13: We appreciate the feedback and have changed this example and also other words throughout the text.

R1.14: The CFE development and characterisation pipeline is omitted from the abstract, it could be included to avoid potentially interested parties overlooking the paper. Furthermore, the paper presents a wealth of data on function of promoter sequences in different species, by making this clear in the abstract the sequence information may be usable in future by parties working with the species characterised

A1.14: We thank the reviewer for the suggestion. We opted not to emphasize the pipeline in the abstract in line with another reviewer's comment (**R2.2**). However, we have added text (line 35-37) to emphasize the latter point about the utility of the data generated here for tuning gene expression in diverse bacteria.

R1.15: L83: Typo, should read "chassis"

A1.15: We have fixed this typo.

R1.16: L89, 96, 184: Throughout the document "new" should be clarified or replaced with something along the lines of "organisms from which a cell-free lysate has not previously been prepared" to refer to microbes which haven't had a cell-free lysate prepared before; the microbes are not new.

A1.16: We appreciate the useful feedback and have made changes with more precise language to address this concern.

R1.17: L105: "support single-channel of measurement" -> "support single-channel measurements"

A1.17: We have now clarified this sentence (line 104-105).

R1.18: L109: You talk about assumptions in multiplexing but neglect the fact that every sequence must be distinguishable, e.g. via a barcode. This should be added.

A1.18: We have edited this sentence to provide this important clarification (line 109-111).

R1.19: L110: Some readers may misunderstand species as in different species of organism, perhaps clarify that nucleic acid species is being referred to.

A1.19: We have modified this sentence to avoid confusion (line 109-111).

R1.20: L132: Include reference to Supplementary Figure 2 which shows the similarity.

A1.20: We have added this reference and thank the reviewer for catching this.

R1.21: L136: You should be explicit about why obtaining uniform numbers of DNA reads across the library helps improve multiplexed measurements. This sort of omission occurs throughout. The authors should carefully check that they backup their statements with proper explanations so that it is not assumed the reader understands their rationale.

A1.21: We appreciate the reviewer's useful feedback and have made modifications to improve clarity. (line 147-149)

R1.22: L138: You don't explain how the TSS was found. A short description would help greatly in understanding the process, with the main details in the methods.

A1.22: This has now been revised in the text (line 151-153, 508-516).

R1.23: L150: "several non-*E. coli* species" -> "several species other than *E. coli*".

A1.23: We have fixed this.

R1.24: L156: Refer the reader to SI figure 4c for the broccoli standard curve.

A1.24: We have fixed this.

R1.25: L159-164: Indicate Figure 3a which shows unrooted phylogenetic tree of species.

A1.25: We have revised the text to cite this figure.

R1.26: L230, 247: References supporting these claims need to be added.

A1.26: We have revised the first sentence and added a citation for the second.

R1.27: L241: I was shocked that the biggest study to date in using sequencing to assess mechanistic aspects of gene regulation (translation initiation and elongation) was not cited, i.e. Cambray et al. Nat Biotech, 2018. Is there a reason for this?

A1.27: This was a very recent paper that is focused on translation, not transcription. However, we share the reviewer's enthusiasm for the paper and have now cited it in the revised text.

R1.28: L291: "increased regulatory dissimilarity"? Consider rewording.

A1.28: We have replaced the language in question with more a more direct description of the data: "increased dissimilarity of transcriptional profiles" (line 323).

R1.29: L324: If the authors would like to introduce potential applications with genetic circuits it is essential that they outline the major issues/challenges using these in bulk reactions. Many circuits with intricate dynamics rely on turn-over of mRNAs and proteins that may not happen in vitro. Such issues and potential solutions should be mentioned.

A1.29: The reviewer is correct that it might be challenging for cell-free reactions to recapitulate long-term *in vivo* dynamics of genetic circuits relying on multiple components turnover from degradation or dilution from cell division. However, recent advances in cell-free methodologies such as the use of degradation tags (Niederholtmeyer et al., 2015) or continuous exchange of reaction contents (dialysis, microfluidics, etc.; Swank et al., 2019, Garamella et al., 2016,) may provide a better mimic of *in vivo* processes while maintaining the simplicity of the *in vitro* format. As the reviewer suggested, we have edited the text to include these limitations and potential solutions (line 352-358). We thank the reviewer for the useful point.

References

- Niederholtmeyer et al., Rapid cell-free forward engineering of novel genetic ring oscillators. *eLife* 4, e09771, (2015).
- Swank et al., Cell-free gene-regulatory network engineering with synthetic transcription factors. *Proc Natl Acad Sci USA* 116, 5892-5901, (2019).
- Garamella et al., The all *E. coli* TX-TL toolbox 2.0: a platform for cell-free synthetic biology. *5*, 344-355, (2016).

R1.30: L337: Rudimentary details of regulatory element mining process would be useful for the reader.

A1.30: We have added details to both the Methods and Results sections.

R1.31: L350: There are no details of how the temperature for each strain was selected for growth. This should be explained in more detail.

A1.31: Host-associated strains were grown at 37°C, while all others were grown at the temperature recommended by the supplier (ATCC, DSMZ) or previous studies (Zhu et al., 2008). We have added the information in the Methods (line 398-400).

References

- Zhu et al., Fermentative hydrogen production by the new marine *Pantoea agglomerans* isolated from the mangrove sludge. *Int J Hydrogen Energy* 33, 6116-6123, (2008)

R1.32: L357: "4,300xg" -> "4,300 x g"

A1.32: We have fixed this.

R1.33: L451: log10 should use subscript.

A1.33: We have fixed this.

R1.34: L452: It would be useful to show the distributions of expression levels and assess that they are normally distributed (a necessary condition for using Z-scores).

A1.34: Appendix Fig S7a shows a boxplot which conveys the approximate normality of expression level distributions.

R1.35: L486: Was any cross-validation done when carrying out the LinearRegression function. If not, why?

A1.35: Cross-validation was performed by using randomly selected 5-80% of the data for bootstrapped training, then testing the model on the left-out (unseen by the model for training) 20-95% of the data (Fig 4b). Example models trained with 10% of the data and tested on 90% of the data are also displayed in Fig 4c. We have revised the text to better describe this process in the Method section.

R1.36: Figure 1: x = y lines could be added to panels b and d, plus the use of a density plot would help better visualize how dense the central regions are.

A1.36: We have added y=x lines in Fig 1b,d and density information to Fig 1d as requested.

R1.37: SI Figure 5: 95% confidence interval lines along line of best fit would be beneficial.

A1.37: This has now been added.

I very much enjoyed reading the paper, believe it is an important methodology, and hope these comments are useful to the authors in improving their manuscript.

We thank the reviewer for their very useful and thorough feedback and enthusiasm about this work.

Reviewer #2:

The authors present a multiplexed, cell-free approach to characterizing DNA regulatory elements using Illumina sequencing. Specifically, their approach uses cell-free transcription and translation with pooled DNA containing the elements to be tested. Then, they reverse transcribe RNA produced in this reaction to obtain cDNA which is then sequenced to obtain abundance of transcripts from each DNA element. This approach is termed DRAFTS. They validate this approach by comparing their results from a DNA library of regulatory elements with a cellular dataset from a previous study. These data align well. The authors then make lysates out of 10 strains of bacteria from evolutionary diverse backgrounds, many of which no previous report of lysate preparation has been published. They perform a simple magnesium and potassium optimization of each lysate using transcription as an output. They take these 10 lysates and test a library of 421 regulatory sequences through DRAFTS and compare their transcriptional outputs in relation to their evolutionary distance from one another. They also find GC-content of these regulatory sequence to be important. They then use this data to generate a somewhat predictive model to describe transcription in each lysate. They use that model to predict transcription activation for 7,003 sequences and use DRAFTS to test these predictions. Their models are roughly predictive ($r=0.52-0.71$). Then, they mix lysates from divergent species together and show that by varying the ratio of each species' lysate they can modulate transcription to be more or less like each individual strain suggesting that the impact of each species might be additive. This work represents an important step forward for cell-free systems in the context of prototyping in that it brings together next generation sequencing and multiplexed reactions. This is the biggest innovation and sets an important precedent. There are clear applications toward prototyping DNA elements for in vivo production of transcripts and proteins in a diverse array of bacteria though this work did not do a sufficient job in comparing/showing this. This work will be of interest to the synthetic biology community, is timely, and well written. However, some elements lessened my enthusiasm and those significant concerns are described below.

Major comments

R2.1: While the work is impressive, the work lacks sufficient comparison to cellular data. The authors state that one of the main problems they are trying to address is the lack of connection and understanding between in vitro measurements and in vivo conditions. I am not sure if this paper changes our understanding since the focus is on *E. coli*. While the cell-free experiments in *E. coli* lysates align nicely with *E. coli* cells, which is expected based on previously published work, there is no in vivo data shown for any of the other nine species. This needs to be done to make their claims and to enhance the impact of the paper.

A2.1: In Appendix Fig S5, we show *in vitro* vs *in vivo* comparisons in seven bacterial species, except the isolated strains (*E. fergusonii* and *P. agglomerans*) without established genetic tools yet and *L. lactis* strain which is recalcitrant to large-scale transformation likely due to its strong restriction-modification system demonstrated in Appendix Fig S6. We have revised the text to clarify this point (line 208-212). These comparisons highlight that *in vitro* and *in vivo* measurements have good correlation.

R2.2: The idea that the authors have a unique or novel approach to lysate prep is not supported. Their development of cell-free expression systems of diverse bacterial species appears to follow within slight variations of previously published methods of optimization, including works from the authors (Noireaux) and others such as ref 18, ref 36, ref 39. The authors should put their work in better and more accurate context since their pipeline is just a variation of what has been reported before.

A2.2: The reviewer is correct that our lysate preparation and optimization approach is indeed a combination of several previous publications, which we have cited in the manuscript thoroughly throughout. Our purpose of highlighting the pipeline was not to suggest the steps in the protocol are novel but rather to demonstrate its use across multiple species. Given that

this study represents the largest scale comparison of different cell-free expression systems and is central to how we generated our data, we felt it was important to highlight the results in this section. However, to address the reviewer's concerns, we have edited the text (line 164-165, 167) to clearly state that our pipeline is a combination of approaches from previous studies.

R2.3: Can the authors comment more on why the *L. Lactis*, *B. subtilis*, and *S. enterica* extracts didn't work well? Or that *Lactobacillus reuteri*, *L. plantarum*, and *Bacteroides thetaiotaomicron* didn't? This would be very informative for the field since many cell-free systems have been recently reported from bacterial species using slight variations of the lysate preparation methods used in this paper and those previously published. Also, did they try and not report broader salt conditions to try to get it to work?

A2.3: There are several reasons why *L. lactis*, *B. subtilis*, and *S. enterica* were suboptimal. Restriction enzyme degradation of library DNA likely impacted measurements in *L. lactis* (**Appendix Fig S6**). Additionally, the stains may have endogenous proteases which complicate lysate generation and would require removal through strain engineering to overcome (Kelwick et al., 2016). *L. lactis* also has low intracellular pH and fermentative metabolism, which may require further optimizations of buffers and energy sources. Preparing lysates from Lactobacilli like *L. reuteri* and *L. plantarum* would have similar challenges and we additionally struggled with developing good lysis protocols. Lastly, for anaerobic microbes such as *Bacteroides thetaiotaomicron*, which we have tested in a very limited setting unsuccessfully, it is possible that anaerobic-optimized energy sources and conditions are needed as well as the use of alternative promoters required by these *Bacteroides* species (Whitaker et al., 2017). In regards to salt concentrations, we only tested the ranges displayed in **Fig 2**, which were derived from previous studies, although it is possible that testing a broader range could be helpful for some species (Moore et al., 2017). Based on the reviewer's comment, we have revised the text in the Discussion section to suggest these potential solutions to produce active cell-free systems from diverse organisms (line 342-347).

References

- Kelwick et al., Development of a *Bacillus subtilis* cell-free transcription-translation system for prototyping regulatory elements. *Metab Eng* 38, 370-381, (2016).
- Whitaker et al., Tunable expression tools enable single-cell strain distinction in the gut microbiome. *Cell* 169, 538-546, (2017).
- Moore et al., *Streptomyces venezuelae* TX-TL – a next generation cell-free synthetic biology tool. *Biotechnol J* 12, 1600678 (2017).

R2.4: The comparative characterization of transcription in diverse bacterial lysates is observational and it is unclear what new knowledge is gained by this. For publication in a journal like MSB, rather than a specialty journal, it seems that there should be new insights gained.

A2.4: We would like to stress several points about the significance of our manuscript. 1) Our data represents a rich resource of transcriptional regulatory parts for the synthetic biology community that can be used for constructing genetic circuits in diverse species *in vivo*. 2) Beyond making the measurements, we also provide a modeling framework to assess promoter strength in these diverse species. Note that beyond our prior *in vivo* study (Johns et al 2018, mostly in 3 species), there is a lack of studies attempting to assess regulatory element function for arbitrary sequences in diverse bacteria. 3) As we expand genetic engineering and synthetic biology to a broader range of organisms, the extent to which genetic components such as promoters can be reliably reused in non-model species contexts has remained unclear. Despite the conservation of RNA polymerase and primary sigma factor proteins, there are many examples of common genetic parts developed for *E. coli* having altered behaviors or lacking function altogether when tested in other species (Cardinale and Arkin, 2012). Our

detailed characterization of a decay in regulatory element strength similarity over phylogenetic distance explains these discrepancies and provides a better basis for regulatory element selection in diverse organisms. Furthermore, it has been requested that we change the manuscript type from “Article” to “Method” to emphasize on our new method, which has a clear potential to generate additional important discoveries beyond the applications we presented in this paper.

References

- Johns et al., Metagenomic mining of regulatory elements enables programmable species-selective gene expression. *Nat Methods* 15, 323-329, (2018)
- Cardinale and Arkin, Contextualizing context for synthetic biology – identifying causes of failure of synthetic biological systems. *Biotechnol J* 7, 856-866, (2012).

R2.5: It is unclear how their predictive models differ from previously published regulatory sequence calculators, and whether their prediction are better than the state-of-the-art (<http://salislab.net>). The authors should quantitatively demonstrate that their method is on par or better than the Salis method and comment on this.

A2.5: We would like to emphasize that there are no widely-used tools for predicting promoter strength from arbitrary sequences in bacteria. While there has been some work on modeling approaches (Rhodius and Mutalik, 2010; Urtecho et al., 2019) in *E. coli*, these have not been applied to libraries as large or compositionally diverse as those used in our study, nor have they been applied to other species. The reviewer mentions the pioneering work by Dr. Howard Salis’ group at Penn State University, however, the models developed by the Salis lab (ribosome binding site calculator and others) are for translation initiation, while our work involves the assessment of transcription activation. Linear regression models have been useful in other studies of gene expression in other organisms and our modeling approach was inspired by these studies (Dvir et al., 2013). We have clarified these points in the revised text to better contextualize our results (line 250-253).

References

- Rhodius and Mutalik, Predicting strength and function for promoters of the *Escherichia coli* alternative sigma factor, sigmaE. *Proc Natl Acad Sci USA* 107, 2854-2859, (2010).
- Urtecho et al., Systematic dissection of sequence elements controlling sigma70 promoters using a genomically encoded multiplexed reporter assay in *E. coli*. *Biochemistry* 58, 1539-1551, (2019).
- Dvir et al., Deciphering the rules by which 5'-UTR sequences affect protein expression in yeast. *Proc Natl Acad Sci USA* 110, E-2792-E2801, (2013).

R2.6: In addition, the hybrid cell-free systems is a nice proof-of-concept but the importance or even understand gained from this is unclear. A more thorough analysis of systems biology would be helpful. Also with respect to the hybrid systems, I am still trying to understand the justification of "Effectively harnessing the biochemical capabilities of hybrid lysates will require knowledge regarding how promoters and translation initiation sequences function in the presence of regulatory machineries from constituent species." It seems this wouldn't be required if you just used enzymes from the cells produced before making the lysates. Can the authors better explain the need for the non-expert?

A2.6: We expect the knowledge regarding how promoters and translation initiation sequences function in the presence of regulatory machineries from constituent species would enable more thorough use of the hybrid lysates by taking control over gene expression in mixed context for many different applications including study of cell biology (Salehi-Reyhani et al., 2017), protein synthesis (Zarate et al., 2010), directed evolution (Miller et al., 2006), and also biosensing (Bally et al., 2010; Adamala et al., 2017), and *in vivo* diagnostics and *in situ* drug

delivery (Noireaux and Libchaber, 2004) potentially as artificial minimal cell format with features from multiple organisms without potential risks from using living cells, beyond merely using their mixed metabolic pathways (or enzymes) for metabolite production (Yi et al., 2018). Furthermore, through DRAFTS, we systematically characterized transcriptional capacities of hybrid lysates and revealed that machineries for gene expression from evolutionarily distinct species do not negatively interfere in mixed solution *in vitro* (**Fig 5, Appendix Fig S12,S13**) and species-selectivity of regulatory sequences can be tuned based on the mixing ratios (**Appendix Fig S14**) which could be useful to control many genes in the lysates without extensive engineering of regulatory elements. Taking the reviewer's comment and suggestion into account, we have revised the text to include these discussions and clarify the points (line 277-284, 305-309).

References

- Salehi-Reyhani et al., Artificial cell mimics as simplified models for the study of cell biology. *Exp Biol Med* 242, 1309-1317, (2017).
- Zarate et al., Development of high-yield autofluorescent protein microarrays using hybrid cell-free expression with combined *Escherichia coli* S30 and wheat germ extracts. *Proteome Sci* 8, 32, (2010).
- Miller et al., Directed evolution by *in vitro* compartmentalization. *Nat Methods* 3, 561-570, (2006).
- Bally et al., Liposome and lipid bilayer arrays towards biosensing applications. *Small* 6, 2481-2497, (2010).
- Adamala et al., Engineering genetic circuit interactions within and between synthetic minimal cells. *Nat Chem* 9, 431-439, (2017).
- Noireaux and Libchaber, A vesicle bioreactor as a step toward an artificial cell assembly. *Proc Natl Acad Sci USA* 101, 17669-17674, (2004).
- Yi et al., Synthesis of (R,R)-2,3-butanediol from starch in a hybrid cell-free reaction system. *J Ind Eng Chem* 67, 231-235, (2018).

Minor comments

R2.7: Line 48, isn't cell membrane a better term than cell envelope? I wasn't sure technically speaking what an envelope is.

A2.8: Cell envelope is the proper term for the entire barrier surrounding a bacterial cell, which will include the peptidoglycan cell wall as well as the double membrane in the case of gram-negative species.

R2.9: In supplemental figure 5, only 7 lysates are tested with no comment or justification. Why didn't the authors test the 234 member library in all 10 bacterial strains? This needs to be better described.

A2.9: As described in the answer **A2.1**, we didn't include isolated strains (*E. fergusonii* and *P. agglomerans*), which does not yet have established genetic tools, and *L. lactis* strain, which is recalcitrant to large-scale transformation likely due to its strong restriction-modification system demonstrated in **Appendix Fig S6**. In addition, *E. fergusonii* and *P. agglomerans* were late additions to this work and were included in other figures mainly to demonstrate usefulness of our approach for rapid transcriptional characterizations for freshly isolated mammalian gut and soil strains, and based on our *in vitro* data they appear to function similarly to other gram-negative strains. Also, while *L. lactis* was not included because we have so far failed to transform it at library scale, we would like to stress that the *L. lactis in vitro* transcriptional profiles clustered with the other gram-positive species as one might expect from its phylogeny. To clarify this point, we have added a sentence to the revised text (line 208-212).

R2.10: Line 210, the hypothesis here is that a specific RE, ScrFIR, is present in *L. lactis*. This is a testable hypothesis. It would strengthen the manuscript to test this hypothesis experimentally, validate the claim, and close the loop.

A2.10: Taking the reviewer's suggestion, we examined the stability of a DNA fragment containing a putative recognition site (CCTGG) and quantified its degradation compared to a control DNA fragment (the same sequence but without the motif) in *L. lactis* lysate using qPCR and have added the result as **Appendix Fig S6e**.

R2.11: In supplemental figure 10 what are the units on the y-axis? The authors should convert to microM or nanoM.

A2.11: We agree with the reviewer that this was confusing and have now changed the axes in the **Appendix Fig S12,S13** to Fluorescence, AU, log₂. Details for our fluorescence measurements can be found in the Method section.

R2.12: Lines 146-186, the authors should tone down the language of this being a new pipeline of making lysates. The pipeline presented in this manuscript matches previous reports of lysate preparation and optimization and is not new.

A2.12: We agree to the reviewer that our lysate preparation pipeline is a hybrid of several previously published approaches. To address the reviewer's point, we have modified our language and clearly stated that our pipeline is a combination of approaches used by previous studies in the revised text (line 164-165, 167).

R2.13: Line 180, is your detection sensitive enough to say that your *B. subtilis* and *S. enterica* have any protein produced?

A2.13: As described in the **Appendix Fig S4b**, the raw fluorescence signals acquired from *S. enterica* and *B. subtilis* lysates were low, but sufficiently strong enough to be distinguished from those from negative control reactions (which were subtracted to quantify GFP concentration) based on three biological replicate experiments. While the focus of this manuscript was on transcription, future studies could seek to improve translation yields in these particular organisms.

We thank the reviewer for their time and very useful feedback.

Reviewer #3:

The authors develop and optimize TX-TL assays using the cell lysates from a diverse collection of 10 bacterial strains that support suitably high levels of transcription. They combine oligopool-based library cloning and next-generation sequencing (DNA-Seq, RNA-Seq) to characterize the transcription rates from hundreds to thousands of natural promoters in TX-TL across the 10 species' lysates. From this data, they identify several promoter sequence features that correlate well with higher transcription rates, including -10 & -35 hexamer sequences, GC content, and spacer length. They also re-confirm that AT-rich promoters are generally transcribed better across many organisms, supporting a finding of the authors' previous work. Overall, the key novelty of the work is the development of TX-TL assays from the cell lysates of diverse organisms, and their utilization to measure promoter transcription rates. However, the data analysis overall was fairly underwhelming; we do not learn a great more from the authors' analysis, compared to what is already known in the literature about promoter specificity.

Major comments

R3.1: The authors use the phrase "predictively modeled" or "predictive modeling" in several places in the abstract and manuscript. However, I see only statistical linear regression models that attempt to find correlative relationships between selected features and outcomes, using standard cross-validation techniques that split one data-set into training and test sets. These are not considered predictions in normal usage of the word (ie, forward-engineering) on unseen data. Moreover, the authors do not test the error of what they call a "prediction", for example, by directly comparing a predicted to measured transcription rate. All metrics are global and statistical (e.g. Pearson's R), and it is well-known that Pearson's R metric has several known defects, which make it unsuitable for testing the "predictive" accuracy of any model by itself. Judging from their analysis, I don't think the authors have any intention that their model would be used to predict a promoter's transcription rate across organisms, and yet they use the word "predictive modeling" several times. The authors either need to remove any mention of "predictive modeling" from the manuscript or substantially support their claims by using their model to predict the transcription rates of a diverse collection of new (perhaps synthetic) promoters, followed by construction and TX-TL characterization across their selected organisms. It's important here to see the full error distribution of any model that attempts to make predictions.

A3.1: We thank the reviewer for their detailed feedback. We would like to note that our transcription models were trained by randomly selecting 10% of the data for bootstrapped training, then tested the models on the left-out 90% of the data ("unseen" by the model for training, therefore "new" collection of promoters for the model) (**Fig 4c**). We also repeated this random sub-sampling to train and test the models for 10 times with many different proportions of training and test datasets for thorough cross-validation of the models and displayed the distribution of the results in the **Fig 4b**. Given our rigorous model validation steps, we feel that we have adequately addressed the question about model fidelity. However, we have further evaluated the models trained on RS7003 library data by testing on a separate collection of regulatory sequences (RS1383) and added the results in a new **Appendix Fig S11**. In addition, we have revised the manuscript to eliminate the mention of "predictive modeling" as suggested by the reviewer. We do not imply that our models are predictive but rather that they are useful for assessing different features/signatures of transcriptional activation (e.g. sigma70 strength, -35/-10 motif, GC content). Thus, we have included more details of the analysis in the text and revised instances of "predictive modeling" within the text to more specific terminology throughout, except for the axis labels in **Fig 4C** because it is standard practice to use "Predicted" and "Observed / Measured" in this context to compare "model output" and "experiment output" for test dataset (Dvir et al., 2013).

To complement Pearson's R metric for model evaluation, we have added the error (Predicted – Measured values) distributions of linear regression models of all 10 bacterial species displayed in **Fig 4C** in a new **Appendix Fig S10** as suggested by the reviewer. The normally distributed errors with 80-98% of predictions within 1-log error window for transcription activities spanning >5 orders of magnitude suggested the fitted models are adequate for the actual data. We revised the text accordingly (line 261-271). We greatly appreciate the reviewer for this valuable comment, which has led to important improvements of the the modeling parts of the work.

References

- Dvir et al., Deciphering the rules by which 5'-UTR sequences affect protein expression in yeast. *Proc Natl Acad Sci USA* 110, E-2792-E2801, (2013).

R3.2: Given the different organisms' differences in TX-TL transcriptional output and distinctly different mRNA dynamics across the organisms, it's not clear how the authors are deriving their "Observed Log10 Tx" values. The authors need to very clearly describe (in the main text) how they extract a single number from their characterized mRNA level dynamics across the different organisms.

A3.2: The reviewer is correct that our lysates are not equally productive in terms of total gene expression yields and dynamics over time. However, we emphasize that despite these differences, the library measurements using deep sequencing as a readout enables accurate measurement of relative (not absolute) expression levels across several orders of magnitude with strong correlation between *in vivo* and *in vitro* measurements (**Fig 1d, Appendix Fig S5**). Importantly, we show in **Appendix Fig S2d** that although total RNA output may change over time within a lysate as shown in **Fig 2b** and **Appendix Fig S4a**, relative levels for individual regulatory sequences stay fairly constant between 30 minutes and 4 hours.

While we measured total gene expression output in **Fig 2b** and **Appendix Fig S4,S12,S13** over time, all library sequencing measurements for cell-free reactions in this study were taken after a 30-minute incubation in cell lysates with library DNA, unless otherwise stated. The formula (RNA abundance / DNA abundance) was used to calculate raw transcription levels, then the whole transcription levels in \log_{10} scale from a reaction were transformed to Z-score for normalization across samples as in many similar studies (Sharon et al., 2012; Vainberg Slutskin et al., 2018; Johns et al., 2018), including our own, which we have cited in the text. However, to clarify these points and prevent any confusion, we have now expanded upon this in our text (line 121-128, 140-144) and the Method section (line 432-506), and revised all the related Figure captions (line 811, 837, 852-853, 862-863) to explicitly state this transformation as suggested by the Reviewer. Per a request from the editor, we have also changed our manuscript type from “Article” to “Method” which has involved more thorough detailing of our methods.

References

- Sharon et al., Inferring gene regulatory logic from high-throughput measurements of thousands of systematically designed promoters. *Nat Biotechnol* 30, 521-530, (2012).
- Vainberg Slutskin et al., Unraveling the determinants of microRNA mediated regulation using a massively parallel reporter assay. *Nat Commun* 9, 529, (2018).
- Johns et al., Metagenomic mining of regulatory elements enables programmable species-selective gene expression. *Nat Methods* 15, 323-329, (2018).

R3.3: In Figure 1C, the y-axis is labeled 'Density' (ie, probability density). However, it's clear that the probability distributions shown are not correctly normalized from their raw counts. If a correct probability distribution is integrated across its interval, it must sum to one, which is not true here. To be more clear and correct, the authors should label the y-axis as 'Probability' and divide all densities such that their sum across the interval is equal to one. This will alter the heights of the probability distributions, but not their shapes, which means the authors' conclusions remain supported.

A3.3: We have now changed this figure to contain histograms with discrete probability distribution rather than the density plots with continuous probability distribution and labeled the y-axis label as “Fraction.”

R3.4: Related to Figure 2b, the authors mention several possibilities for why the dynamics of mRNA levels vary across TX-TL reactions made using different organism cell lysates. However, the authors do not mention if the organisms could have been grown in conditions where their sigma factor levels are distinctly different. For example, how do we know if the sigma70 vs. sigmaS (homolog) levels in *S. enterica* cell lysate are the same as in the *E. coli* lysate? These differences would have a large effect on promoter sequence specificity in TX-TL reactions. What steps did the authors take to “regularize” the growth conditions across all organisms so that their sigma factor levels are the same, keeping in mind that exponential growth conditions in one organism are distinct from another?

A3.4: In order to minimize confounding effects from media conditions or growth differences, all of our lysates were prepared from cells growing in rich media at mid-exponential phase (**Appendix Table S1**). Growth rates, including those for strains that did not produce highly

active lysates, appeared to be around the maximum reported in previous studies. This suggests that the cells were not stressed and would likely not have high activity of alternative sigma factors, which typically regulate nutrient limitation and stress response phenotypes. Newly added **Appendix Fig S8**, which shows strong correlation between similarities of transcription profiles and primary sigma factor RpoD amino acid sequences between 10 bacterial species, also suggests the primary sigma factors play the main role in activating transcription in the lysates and supports our claim that more phylogenetically related species share a more similar transcription profile. While we cannot completely rule out the activity of alternative regulators, any changes in gene expression that would result from their activities would also be observed in analogous *in vivo* experiments and are unavoidable when comparing organisms that are inherently different from one another. In addition, from a practical standpoint, most synthetic biology studies are performed in similar rich media and at mid-exponential phase and we believe our data is relevant to such conditions. To clarify these points, we have revised the Method section to include these considerations (line 437-440).

We thank the reviewer for their time and constructive feedback which helped improving the manuscript significantly.

Thank you for sending us your revised manuscript. We have now heard back from the three reviewers who were asked to evaluate your study. As you will see the reviewers are satisfied with the modifications made and think that the study is now suitable for publication.

Before we formally accept your manuscript, we would ask you to address a few remaining editorial issues listed below.

REFEREE REPORTS

Reviewer #1:

The authors have adequately addressed all of my comments.

Reviewer #2:

The authors provided a lot of new text and detailed responses to my questions. I think the manuscript is now suitable for publication.

Reviewer #3:

The authors have responded well to this reviewer's comments, particularly the expanded modeling & analysis of their results, and have addressed this reviewer's concerns.

Corresponding Author Name: Harris H. Wang

Manuscript Number: MSB-19-8875R